# Evolutionary stability of collateral sensitivity to antibiotics in the model pathogen *Pseudomonas aeruginosa*

Camilo Barbosa[1†], Roderich Römhild[1,2†‡], Philip Rosenstiel[3], Hinrich Schulenburg[1,2]*

[1]Department of Evolutionary Ecology and Genetics, University of Kiel, Kiel, Germany; [2]Max Planck Institute for Evolutionary Biology, Plön, Germany; [3]Institute of Clinical Molecular Biology, UKSH, Kiel, Germany

**Abstract** Evolution is at the core of the impending antibiotic crisis. Sustainable therapy must thus account for the adaptive potential of pathogens. One option is to exploit evolutionary trade-offs, like collateral sensitivity, where evolved resistance to one antibiotic causes hypersensitivity to another one. To date, the evolutionary stability and thus clinical utility of this trade-off is unclear. We performed a critical experimental test on this key requirement, using evolution experiments with *Pseudomonas aeruginosa*, and identified three main outcomes: (i) bacteria commonly failed to counter hypersensitivity and went extinct; (ii) hypersensitivity sometimes converted into multidrug resistance; and (iii) resistance gains frequently caused re-sensitization to the previous drug, thereby maintaining the trade-off. Drug order affected the evolutionary outcome, most likely due to variation in the effect size of collateral sensitivity, epistasis among adaptive mutations, and fitness costs. Our finding of robust genetic trade-offs and drug-order effects can guide design of evolution-informed antibiotic therapy.

**\*For correspondence:**
hschulenburg@zoologie.uni-kiel.
de

†These authors contributed
equally to this work

**Present address:** ‡Department
of Medical Biochemistry and
Microbiology, Uppsala
University, Uppsala, Sweden

**Competing interests:** The
authors declare that no
competing interests exist.

**Reviewing editor:** Csaba Pal,
Biological Research Centre of the
Hungarian Academy of Sciences,
Hungary

## Introduction

Treatment of infectious diseases and cancer often fail because of the rapid evolution of drug resistance (*Bloemberg et al., 2015*; *Davies and Davies, 2010*; *Gottesman, 2002*; *Zaretsky et al., 2016*). Optimal therapy should thus anticipate how resistance to treatment evolves and exploit this knowledge to improve therapy (*Gatenby et al., 2009*; *Imamovic and Sommer, 2013*). One promising strategy is based on evolved collateral sensitivity: the evolution of resistance against one drug A concomitantly causes hypersensitivity (i.e., collateral sensitivity) to a second drug B (*Szybalski and Bryson, 1952*). If evolved collateral sensitivity is reciprocal, it can – in theory – trap the bacteria in a double bind, thereby preventing the emergence of multidrug resistance during treatment (*Baym et al., 2016*; *Pál et al., 2015*; *Roemhild and Schulenburg, 2019*). Recent large-scale studies have demonstrated that evolved collateral sensitivity is pervasive in laboratory strains and clinical isolates of distinct bacterial species (*Barbosa et al., 2017*; *Imamovic et al., 2018*; *Imamovic and Sommer, 2013*; *Jansen et al., 2016*; *Jiao et al., 2016*; *Lázár et al., 2014*; *Lázár et al., 2013*; *Oz et al., 2014*; *Podnecky et al., 2018*) as well as cancer cells (*Dhawan et al., 2017*; *Pluchino et al., 2012*; *Shaw et al., 2016*; *Zhao et al., 2016*). More importantly, evolved collateral sensitivity can slow down resistance evolution during combination (*Barbosa et al., 2018*; *Rodriguez de Evgrafov et al., 2015*; *Munck et al., 2014*) and sequential therapy (*Kim et al., 2014*; *Roemhild et al., 2015*), and also limit the spread of plasmid-borne resistance genes (*Rosenkilde et al., 2019*).

Although collateral sensitivity appears to be pervasive, its utility for medical application is still dependent on several additional factors. Firstly, the evolution of collateral sensitivity should ideally

**eLife digest** Over time bacteria can undergo a number of genetic mutations that allow them to evolve in response to changes in their surrounding environment. This process of 'bacterial evolution' is one of the major causes of antibiotics resistance, whereby disease-causing microorganisms become resistant to multiple drugs and can no longer be destroyed using antibiotic treatment. However, when bacteria become resistant to a drug this can result in an evolutionary trade-off known as 'collateral sensitivity' – when evolving resistance to one drug causes bacteria to gain increased sensitivity to another.

Now, Barbosa, Roemhild et al. have investigated whether this evolutionary trade-off could be exploited to tackle the antibiotic crisis and prevent bacteria adapting to different treatments. If this evolutionary trade-off is to be used medically, it must be stable long enough for the bacteria population to either become extinct, or less able to evolve multi-drug resistance. To test how stable collateral sensitivity is over time, Barbosa, Roemhild et al. studied the bacterium *Pseudomonas aeruginosa* which is known to evolve collateral sensitivity to certain drug treatments.

*P. aeruginosa* were subjected to two rounds of evolution: first, bacteria were evolved to resist 'Drug A' and at the same time became more sensitive to another drug, 'Drug B'. The bacteria were then allowed to adapt to Drug B either alone or in the presence of Drug A. These evolutionary experiments revealed that the following factors affected the stability of the trade-off: the molecular structure of the antibiotic bacteria evolved sensitivity to, the strength of the original evolutionary trade-off (i.e. how sensitive bacteria became), the order drugs were administrated, and whether resistance came at a large fitness cost (i.e. when the genetic mutations promoting resistance affect bacteria's ability to replicate and survive in normal conditions).

According to the World Health Organization, *P. aeruginosa* is the second most problematic multi-drug resistant bacteria. The data collected in this study could therefore be used to develop a new antibiotic therapeutic strategy for fighting this bacterium, as well as other microbes which are resistant to multiple drugs.

be repeatable for a given set of conditions (**Nichol et al., 2019**). This means that independent populations selected with the same drug should produce identical collateral effects when exposed to a second one. Such high repeatability is not always observed. Recent work even identified evolution of contrasting collateral effects (i.e., some populations with evolved collateral sensitivity and others with cross-resistance) for different bacteria, including *Pseudomonas aeruginosa* (**Barbosa et al., 2017**), *Escherichia coli* (**Nichol et al., 2019**; **Oz et al., 2014**), *Enterococcus faecalis* (**Maltas and Wood, 2019**), and a BCR-ABL leukemeia cell line (**Zhao et al., 2016**). These patterns are likely due to the stochastic nature of mutations combined with alternative evolutionary paths to resistance against the first selective drug, subsequently causing distinct collateral effects against other drugs (**Barbosa et al., 2017**; **Nichol et al., 2019**). Secondly, the evolution of collateral sensitivity should ideally be repeatable across conditions, for example different population sizes. This is not always the case. For example, an antibiotic pair, which consistently produced collateral sensitivity in small *Staphylococcus aureus* populations (e.g., $10^6$), instead produced complete cross-resistance in large populations (e.g., $10^9$) and thus an escape from the evolutionary constraint, most likely due to the higher likelihood of advantageous rare mutations under these conditions (**Jiao et al., 2016**).

A third and largely unexplored factor is that evolved collateral sensitivity and, hence, the resistance trade-off should be stable across time. This implies that bacteria either cannot evolve to overcome collateral sensitivity and thus die out, or, if they achieve resistance to the new drug B, they should concomitantly be re-sensitized to the original drug A. Two recent studies, both with different main research objectives, yielded some insight into this topic. One example was focused on historical contingency during antibiotic resistance evolution of *P. aeruginosa* (**Yen and Papin, 2017**). As part of the results, the authors identified lineages with evolved resistance against ciprofloxacin that simultaneously showed increased sensitivity to piperacillin and tobramycin. The reverse pattern (i.e., evolved high resistance to either piperacillin or tobramycin and increased sensitivity to ciprofloxacin) was not observed and, thus, this case represents an example of uni-directional collateral sensitivity. The subsequent exposure of the ciprofloxacin-resistance lineages to either piperacillin or tobramycin

led to the evolution of resistance against these two antibiotics and substantial (yet not complete) re-sensitization to ciprofloxacin. The second study focused on evolving *E. coli* populations in a mor-bidostat, in which the bacteria were exposed to repeated switches between two drugs (*Yoshida et al., 2017*). The evolution of multidrug resistance was only prevented in the two treat-ments with polymyxin B that were also characterized by evolved collateral sensitivity, although again only in one direction (*Yoshida et al., 2017*). To date, the general relevance of this third factor is still unclear, especially for conditions when collateral sensitivity is reciprocal and when the evolving pop-ulations are also allowed to go extinct (i.e., they cannot overcome the evolutionary trade-off).

Here, we specifically tested the potential of the model pathogen *P. aeruginosa* to escape recipro-cal collateral sensitivity through *de-novo* evolution. We focused on the first switch between two drugs, because the evolutionary dynamics after this first switch will reveal the ability of the bacteria to adapt to the second drug, against which they evolved sensitivity, and, if so, whether this causes re-sensitization to the first drug. These two aspects are key criteria for applicability of a treatment strategy that exploits evolved collateral sensitivities. Our analysis is based on a two-step evolution experiment. Bacteria first evolved resistance against a first drug A and concomitant sensitivity against a second drug B. Thereafter, bacteria were subjected to a second evolution experiment, dur-ing which they were allowed to adapt to the second drug B, either alone or additionally in the pres-ence of the first drug A. Phenotypic characterization of the evolved bacteria was combined with genomic and functional genetic analyses, in order to determine the exact targets of selection under these conditions. We finally validated the findings made by performing further, independent sets of similar two-step evolution experiments.

## Results

The experimental design of our two-step evolution experiment is illustrated in *Figure 1a*. We took advantage of previously evolved, highly resistant *P. aeruginosa* populations, which we obtained from serial passage experiments with increasing concentrations of clinically relevant bactericidal antibiot-ics (drug A, *Figure 1a*) (*Barbosa et al., 2017*). From these, we identified two cases of reciprocal col-lateral sensitivity, including (i) piperacillin/tazobactam (PIT) and streptomycin (STR), or (ii) carbenicillin (CAR) and gentamicin (GEN). In the current study, we now re-assessed the reciprocity of collateral effects using dose-response analysis (*Figure 1b and c*, *Supplementary file 1*–Figure 1–supplementary table 1, *Figure 1—source data 1*). Thereafter, we isolated resistant colonies from these populations and switched treatment to the drug, against which collateral sensitivity was observed (drug B, *Figure 1a*). By starting this treatment step with clonal lineages, we here tested the ability of *P. aeruginosa* to overcome collateral sensitivity by *de-novo* mutation. The evolutionary challenge was initiated at sub-inhibitory concentrations of each drug (vertical black dashed lines in *Figure 1b and c*, *Supplementary file 1*–Figure 1–supplementary table 1, *Figure 1—source data 1*), followed by linear concentration increases at two different rates: mild or strong (vertical orange and blue dashed lines respectively, *Figure 1b and c*, *Supplementary file 1*–Figure 1–supplementary table 1, *Figure 1—source data 1*). We specifically chose linear increases, because our main objective was to better understand the evolutionary dynamics occurring during the first switch of a collateral sensitivity cycling strategy. Linear increases would, in this case, facilitate evolutionary rescue and pro-vide ample opportunity to escape the double bind, thereby yielding a conservative measure for the applicability of collateral sensitivity as a treatment strategy. We additionally considered treatments where antibiotics were also switched to collateral sensitivity, but selection by drug A was continued in combination with drug B; hereby denoted as constrained environments. Overall, four selective conditions were run in parallel: mild or strong increases of the second drug B in either the presence (constrained) or absence (unconstrained) of the first drug A (*Figure 1a*). We further included control experiments without antibiotics. To determine treatment success, we monitored bacterial growth with continuous absorbance measurements, quantified frequencies of population extinction, and characterized changes in antibiotic resistance of the evolved bacteria as previously evaluated for *P. aeruginosa* and other bacteria (*Barbosa et al., 2018*; *Barbosa et al., 2017*; *Hegreness et al., 2008*).

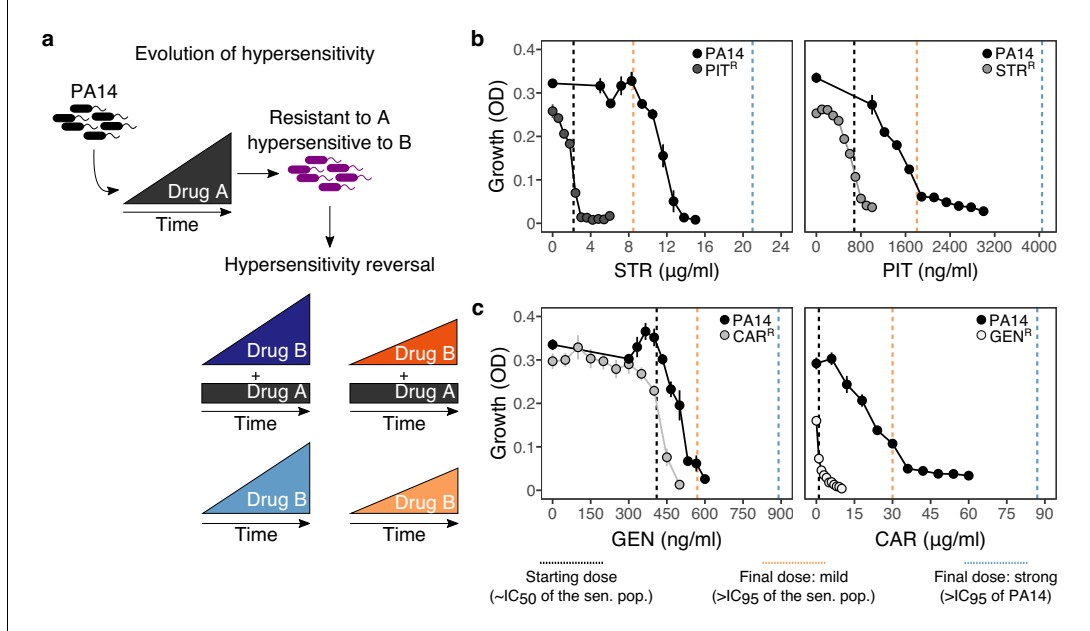

**Figure 1.** Reciprocal collateral sensitivity and experimental design. (a) Two-step experimental evolution: resistant populations of *P. aeruginosa* PA14 were previously experimentally evolved (*Barbosa et al., 2017*) with increasing concentrations of a particular drug (here labeled A), resulting in bacteria becoming hypersensitive to other drugs (here labeled B). In a second step, A-resistant clones were experimentally evolved in the presence of drug B, using four selection regimes: (i) strong dose increase of drug B in the presence of a constant high dose of drug A; (ii) mild dose increase of B in the presence of A; (iii) strong dose increases of B in the absence of A; and (iv) mild dose increase of B in the absence of A. Concentrations of B were increased using linear ramps starting at $IC_{50}$ (dashed black lines) and ending at levels above the $IC_{95}$ of the collaterally sensitive clone (mild increases, dashed orange line), or that of the PA14 wild type strain (strong increases, dashed black lines; detailed information on concentrations in Supplementary Table 1). (b) Validated reciprocity of collateral sensitivity for the isolated resistant clones and drug pair PIT/STR, and (c) CAR/GEN. Mean ±CI95, n = 8. Vertical dashed lines indicate the starting (black) and final doses of the mild (light orange) and strong drug increases (light blue). CAR, carbenicillin; GEN, gentamicin; STR, streptomycin; PIT, piperacillin with tazobactam; superscript R denotes resistance against the particular drug. The following supplementary table and source data are available for *Figure 1b and c*: *Supplementary file 1*-Figure 1-supplementary table 1 and *Figure 1—source data 1*.

The online version of this article includes the following source data for figure 1:

**Source data 1.** Mean optical density and CI95 values obtained after 12 hr of growth in minimal media and different antibiotics as reported in *Figure 1*.

## Extinction rates were high even under mild selection regimes

The imposed antibiotic selection frequently caused population extinction (*Figure 2*, *Figure 2—source data 1*), even though sublethal drug concentrations were used. Extinction events occurred significantly more often when selection for the original resistance was maintained by the presence of both drugs (extinction in constrained treatments *vs.* only one drug, $\chi^2$ = 12.9, *df* = 1, p<0.0001; *Figure 2*). In treatments with only the second drug B, extinction occurred significantly more often under strong, but not mild concentration increases (strong vs. mild increases in unconstrained environments, $\chi^2$ = 5.5, *df* = 1, p=0.019). Drug switches with the antibiotic pair STR/PIT was particularly successful, with 33 extinction events (~51%, *Figure 2*). The results differed for the CAR/GEN pair, which produced only seven extinctions (~10%), all restricted to one drug order, GEN>CAR, suggesting asymmetry in the ability to counter collateral sensitivity. The observed differences in the extinction levels of the pairs considered here are unlikely to be the result of differences in combined synergy, since both combinations (PIT/STR and CAR/GEN) are synergistic against *P. aeruginosa* (*Barbosa et al., 2018*). From this, we conclude that strong genetic constraints against an evolutionary response to collateral sensitivity caused frequent population extinctions for STR/PIT switches, whereas evolutionary rescue was possible for the GEN/CAR pair, although influenced by drug identity and order.

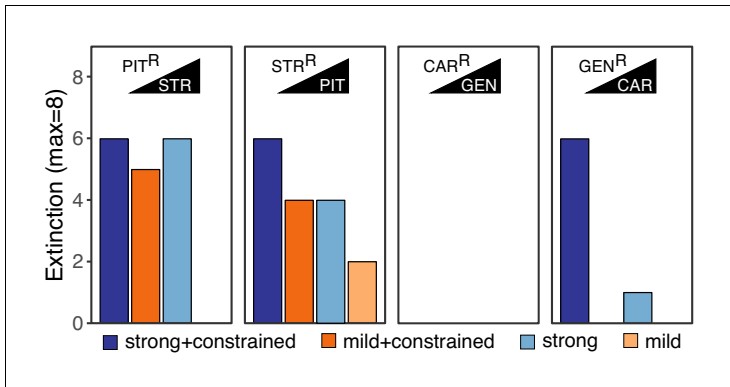

**Figure 2.** Extinction events during second step of experimental evolution. From left to right, extinction events for PIT[R]-clones challenged with STR, STR[R]-clones challenged with PIT, CAR[R]-clones challenged with GEN, and GEN[R]-clones challenged with CAR. Superscript R indicates resistance against the particular antibiotic, as evolved during the first step of the evolution experiment. The following source data is available for *Figure 2*: *Figure 2—source data 1*.

The online version of this article includes the following source data for figure 2:

**Source data 1.** Count data of extinction events as reported in *Figure 2*.

## Novel resistance evolved rapidly in many of the surviving populations

We subsequently focused our analysis of the evolutionary dynamics on the surviving populations of the CAR/GEN pair and identified rapid adaptive responses, especially when not constrained by the presence of the two drugs (*Figure 3*). We measured bacterial adaptation using relative biomass (see Materials and methods and *Roemhild et al., 2018*) and found it to have increased in all surviving populations (*Figure 3a and b*, *Supplementary file 1*–Figure 3–supplementary table 1, *Figure 3— source data 1*). For both drug orders, the increase was significantly slower in the constrained treatments, and, to a lesser extent, for the strong concentration increases (*Figure 3a and b*). Consistent with the asymmetry in extinction, the CAR>GEN switch (with no extinction) maintained a high relative biomass across time, while the reverse direction GEN>CAR (with extinction) produced lower relative biomass levels. These results indicate that *P. aeruginosa* can evolve resistance against a drug, to which it had previously shown hypersensitivity, and that such evolutionary rescue is favored for the suboptimal switch. For the STR/PIT pair, we generally obtained similar results (*Figure 3—figure supplement 1a and b*, *Figure 3—source data 1*); yet because of the few surviving populations and high variation among these rare survivors, the results remained inconclusive. We thus continued to focus on the evolved populations for the CAR/GEN pair and asked how the new adaptation influenced the original drug resistances.

## Drug order determined re-sensitization or emergence of multidrug resistance

Adaptation in the surviving populations of the CAR/GEN pair caused multidrug resistance in the suboptimal switch, but re-sensitization to similar levels to those of the PA14 ancestor (*Figure 3—figure supplement 2*, *Figure 3—figure supplement 2—source datas 1* and *2*) in the alternative switch (*Figure 3c and d*, *Figure 3—source data 2*). In detail, all surviving populations significantly increased resistance against the second drug ($IC_{90}$-fold change between two and >64 times that of the starting populations; *Figure 3c and d*, *Supplementary file 1*–Figure 3–supplementary tables 2 and 3, *Figure 3—source data 2*) – in agreement with the recorded biomass dynamics. In the suboptimal switch, CAR>GEN, all populations maintained their original resistance, thereby yielding bacteria with multidrug resistance. This was different for the alternative direction GEN>CAR, where the original resistance was only maintained when both drugs were present in combination (constrained environments). Only under unconstrained evolution, we observed cases of significant re-sensitization to the first drug. We conclude that drug order can determine treatment efficacy, enhance or minimize multidrug resistance, and, in specific cases, lead to a re-sensitization towards the first drug in

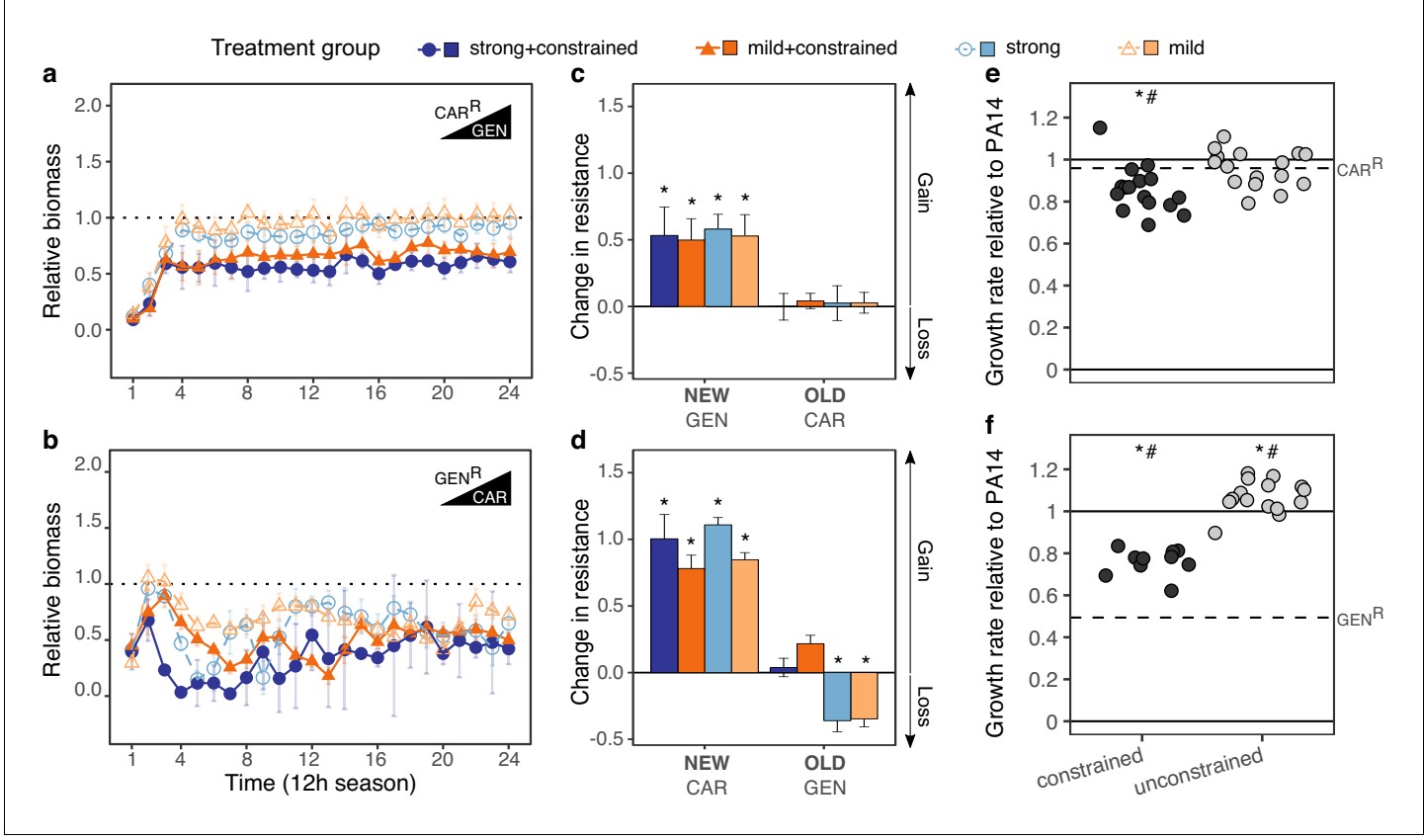

**Figure 3.** Contrasting evolutionary stability of collateral sensitivity for CAR>GEN and GEN>CAR switches. Evolutionary dynamics of surviving populations expressed as relative biomass for (**a**) CAR^R-populations during selection with GEN, and (**b**) GEN^R-populations during selection with CAR. The dotted horizontal line indicates growth equal to untreated controls. Mean ±CI95, number of biological replicates differs due to extinction (n = 2–8). Changes in antibiotic resistance at the end of the second-step evolution experiment for (**c**) CAR^R-populations after selection with GEN and (**d**) GEN^R-populations after selection with CAR. Resistance was tested either against the drug towards which bacteria initially showed resistance after the first evolution experiment (indicated as OLD), or the drug used during the second experiment (indicated as NEW). The change is measured by cumulative differences in dose-response before and after the second evolution experiment (i.e., the original antibiotic resistant clone *versus* its evolved descendants). Mean ±CI95, n = 2–8 biological replicates (differences due to extinction). Asterisks indicate significant changes in resistance (one-sample *t*-test, μ = 0, FDR-adjusted probabilities). Change of maximum exponential growth rate in drug-free conditions for the evolved lineages, relative to wild type PA14 for (**e**) CAR^R>GEN lineages and, (**f**) GEN^R>CAR lineages. This measure is used to explore the presence of a general adaptation trade-off. The evolved lineages were grouped by whether experimental evolution was performed under constrained conditions (i.e., presence of drug A and B; dark gray circles) or not (i.e., only presence of drug B; light gray circles). The dashed line in each panel then indicates relative growth rate of the starting resistant population. Asterisks show significant increases or decreases in growth rate relative to the wild type PA14 (One-sample *t*-test, μ = 1, p<0.004), while numerals indicate significant increases or decreases relative to the starting population (dashed lines in each panel, One-sample *t*-test, μ = GEN^R or CAR^R, p<0.004). Number of populations per group and experiment vary due to extinction (min = 10, max = 16). The following supplementary figures, tables and source data is available for *Figure 3*: *Figure 3—figure supplement 1*, *Figure 3—figure supplement 2*, *Supplementary file 1*-Figure 3-supplementary tables 1-3, and *Figure 3—source data 1*; *Figure 3—source data 2*; *Figure 3—source data 3*.

The online version of this article includes the following source data and figure supplement(s) for figure 3:

**Source data 1.** Evolutionary dynamics summarized by the area under the curve (AUC) across experimental seasons relative to the reference treatment with no drugs.

**Source data 2.** Dose-response curves data of surviving populations challenged with CAR, GEN, STR and PIT.

**Source data 3.** Growth rate estimates of the surviving population challenged with CAR or GEN, as reported in *Figure 3*.

**Figure supplement 1.** Evolutionary stability of collateral sensitivity for PIT^R>STR and STR^R>PIT switches.

**Figure supplement 2.** Re-sensitization to gentamicin (GEN) upon adaptation to carbenicillin (CAR).

**Figure supplement 2—source data 1.** Dose-response curves data of surviving populations adapted to unconstrained environments (strong and mild), as well as the PA14 wild type against GEN.

**Figure supplement 2—source data 2.** Change in resistance of populations adapted to unconstrained environments (strong and mild) relative to the PA14 wild type against GEN.

the surviving populations, as required for applicability of collateral sensitivity cycling (*Imamovic and Sommer, 2013*).

We hypothesized that the contrasting evolutionary outcomes in constrained *versus* unconstrained treatments of the GEN>CAR switch were caused by an additional trade-off, in this case between drug resistance and growth rate. We obtained a proxy for such a general trade-off, which is comparable across the distinct antibiotic treatments, by measuring maximum exponential growth rate under drug-free conditions and standardizing it against the corresponding growth rate for the ancestral PA14 strain. Even though measured in drug-free environments, a possible reduction in growth rates may still indicate a general growth constraint or adaptation trade-off, which is also relevant under other conditions (i.e., under antibiotic exposure). The starting clones for the second evolution experiment indeed showed significantly impaired growth rates under drug-free conditions, with up to 50% reductions relative to the ancestor (*Barbosa et al., 2017*). As a consequence, selection may have favored variants for which both trade-offs (i.e., the collateral sensitivity and also the general adaptation trade-offs) were ameliorated during evolution. For the GEN>CAR switch, we indeed found a significant increase in growth rate relative to the wild type PA14 in the unconstrained treatments (*Figure 3f*, *Figure 3—source data 3*). Constrained populations for this particular switch still showed a significantly reduced growth rate relative to PA14, with however significantly improved values relative to the starting population (*Figure 3f*, *Figure 3—source data 3*). The alternative switch did not show similar variations, mainly due to the fact that the costs of the initial population were not as high (*Figure 3e*, *Figure 3—source data 3*). Altogether, this data suggests that selection under the GEN>CAR unconstrained treatments provided the dual advantage of reversing two previously acquired evolutionary trade-offs, namely hypersensitivity to a second drug and increases in growth rate. Thus, re-sensitization could have been favored over multidrug resistance because of the associated adaptation trade-off that can ultimately influence treatment outcome upon collateral sensitivity switches.

## Whole genome sequencing identified possible targets of antibiotic selection

We used population genomic analysis to characterize specific functional changes that were likely targeted by antibiotic selection and allowed populations to survive the second evolution experiment for the CAR/GEN pair (*Figure 4*, *Figure 4—source data 1*). In particular, we sequenced whole genomes of the resistant starting clones from the beginning and 21 surviving populations from the end of the second evolution experiment. Our results reveal that the evolution of multidrug resistance in the suboptimal switch CAR>GEN can be explained by the sequential fixation of mutations including, under unconstrained conditions, those in *ptsP* (*Figure 4a*, *Figure 4—source data 1*), a main component of the global regulatory system of ABC transporters and other virulence factors (*Feinbaum et al., 2012*). Similarly, under constrained conditions, we found mutations in the NADH-dehydrogenase genes *nuoD or nuoG* (*Figure 4a*, *Figure 4—source data 1*), which are known to influence proton motive force and resistance against aminoglycosides upon inactivation (*El'Garch et al., 2007*).

For the more effective switch GEN>CAR, multidrug resistance in the constrained treatments coincided with mutations in *mexR*, *phoQ, and cpxS,* an independent regulator of MexAB-OprM (*Li et al., 2016*) and two-component regulators involved in aminoglycoside resistance (*Gooderham and Hancock, 2009*) and envelope stress response (*Roemhild et al., 2018*), respectively. The re-sensitization towards the first drug in the unconstrained GEN>CAR treatments was associated with two main types of mutational changes at high frequencies across several replicates, including (i) mutations in *nalC* and *nalD* that upregulate the expression of the multidrug-efflux system MexAB-OprM in *P. aeruginosa* (*Li et al., 2016*); and (ii) large deletions in *pmrB*, which is part of a two-component regulatory system (*Figure 4b*, *Figure 4—source data 1*). Mutations in *nalC* were previously shown to mediate both resistance to CAR and hypersensitivity to GEN (*Barbosa et al., 2017*). Thus, re-sensitization to GEN may be caused by antagonistic pleiotropy of *nalC* mutations that override the resistance of the original *pmrB* mutation (*Figure 4*, *Figure 4—source data 1*). In addition, there may be epistasis between the two functional modules. A complementary mechanism for re-sensitization against GEN is the re-mutation of *pmrB* (*Figure 4*, *Figure 4—source data 1*). In three cases *nalC* mutations coincided with mutations in *pmrB*, including two deletions of 17 and 225 base pairs. Whilst the original SNP in *pmrB* altered gene function, the latter deletions may have

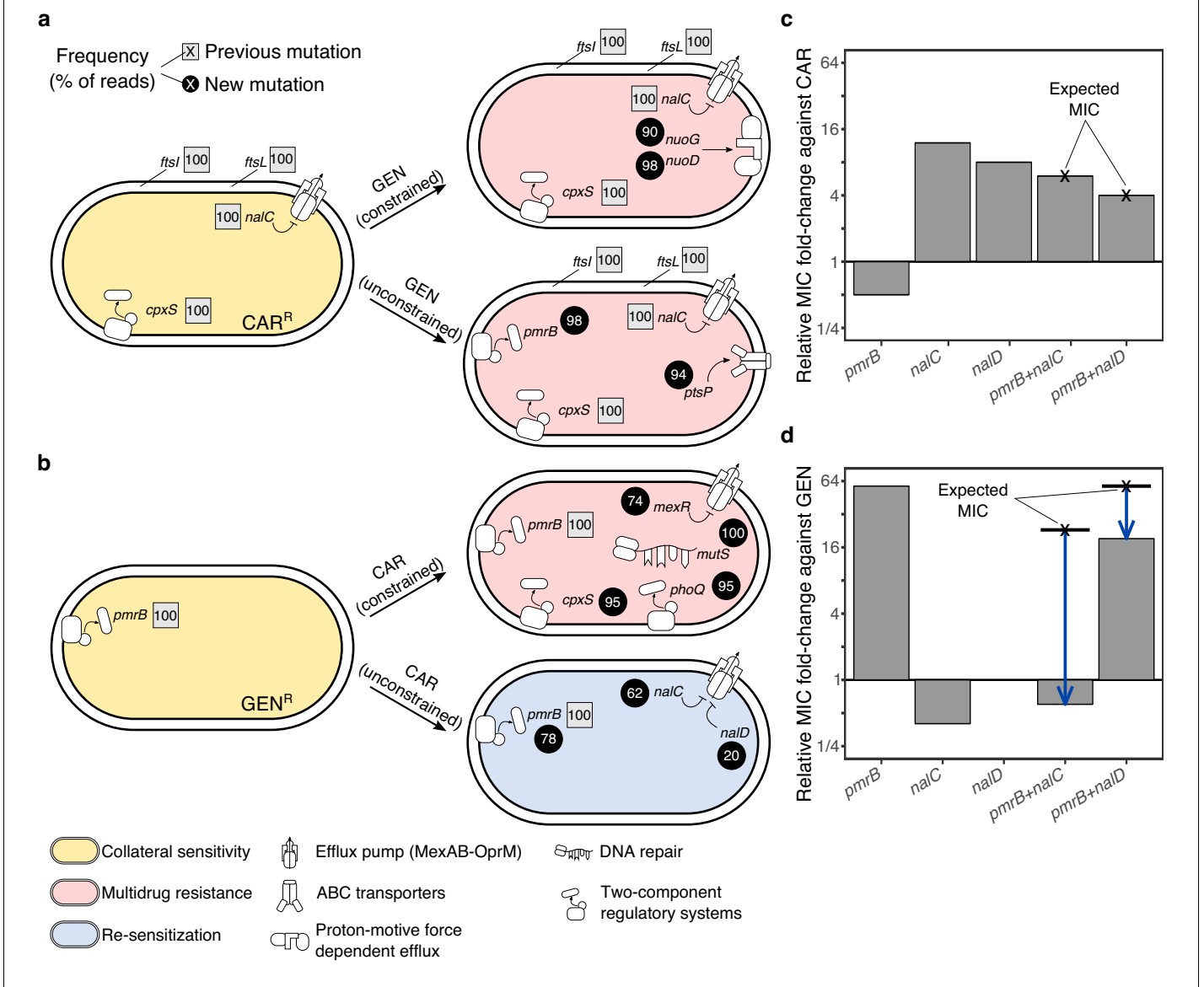

**Figure 4.** Genomics of experimental evolution for the CAR/GEN drug pair. (**a**) Most relevant genomic changes in CAR-resistant populations selected with GEN, and (**b**) GEN-resistant populations selected with CAR. Square symbols next to gene names indicate ancestral resistance mutations (obtained from *Barbosa et al., 2017*), and circles indicate newly acquired mutations. The numbers inside these symbols indicate variant frequencies (as inferred by the percentage of reads in population genomics data) and correspond to the lowest frequency found among the sequenced populations from the respective treatment. The evolved resistance phenotype is highlighted by color shading (see legend in bottom left). (**c**) MIC relative to PA14 against CAR and (**d**) GEN, of single and double mutant strains. The cross and bold horizontal lines indicate the MIC as expected by addition of the individual effects. Blue arrows highlight epistatic effects. The following supplementary source data is available for *Figure 4*: *Figure 4—source data 1*; *Figure 4— source data 2*.

The online version of this article includes the following source data for figure 4:

**Source data 1.** Genetic changes compared to *Pseudomonas aeruginosa* PA14 wild type strain as determined by whole-genome resequencing.
**Source data 2.** MIC values for several constructed mutants against CAR and GEN, as reported in *Figure 4*.

suppressed the expression of the original SNP by pseudogenizing the gene. We conclude that mutations in the *nalC or nalD* regulators of the MexAB-OprM pump, sometimes in combination with follow up mutations in *pmrB* are likely to account for the re-sensitization towards the first drug GEN.

## Functional genetic analysis revealed asymmetric epistasis among adaptive mutations

We next investigated whether epistasis between the two functional modules of efflux regulation (MexAB-OprM regulation by *nalC* or *nalD*) and surface charge modification (*pmrB*) may have contributed to re-sensitization using functional genetic analysis. The respective single and double mutations were re-constructed in the common ancestral background of PA14 (see Materials and methods for the specific mutations) and changes in resistance against CAR and GEN were measured using fold-change of minimal inhibitory concentrations (MIC, *Figure 4c and d*, *Figure 4—source data 2*). On CAR, the *pmrB* mutant had half of the MIC of PA14 (confirming collateral sensitivity), whilst *nalC* and *nalD* mutants had increased resistance to CAR. The double mutants had lower MIC on CAR than the *nalC* and *nalD* single mutants (*Figure 4c*, *Figure 4—source data 2*). The extent of MIC changes in the double mutants corresponded to the product of the individual effects in the respective single mutants, thus indicating an additive interaction among mutations on CAR. On GEN, however, the double mutants had substantially lower MICs than expected from the single mutants (*Figure 4d*, *Figure 4—source data 2*), strongly suggesting negative epistasis. In detail, GEN-resistance relative to PA14 was 0.4x for *nalC* (collateral sensitivity), 1x for *nalD*, and 57x for *pmrB* (*Figure 4d*). The *pmrB*, *nalD* double mutant had 3x lower MIC to GEN than expected from the individual effects. The *pmrB*, *nalC* double mutant had >30 x lower MIC to GEN than expected from the individual effects, resulting in greater sensitivity than PA14 (*Figure 4d*, *Figure 4—source data 2*). Altogether, we conclude that re-sensitization to GEN is mediated by antagonistic pleiotropy and negative epistasis.

## Repetition of experimental evolution revealed an influence of drug type on population extinction and drug re-sensitization

To validate and generalize our findings, we repeated evolution experiments for 14 cases of collateral sensitivity, using a total of 38 distinct resistant populations (*Figure 5a*; *Figure 5—source data 1*), obtained from independent biological replicates of our previous study (*Barbosa et al., 2017*). We initiated experiments with the available populations (*Barbosa et al., 2017*), rather than clones, and applied a less severe bottleneck of $10^7$ cells. Therefore, any adaptive changes may not exclusively rely on *de-novo* mutations but could also result from available genetic diversity, thus reflecting a more general scenario for evolutionary adaptation to collateral sensitivity. The 14 cases of collateral sensitivity included the same four examples tested above and 10 additional cases, encompassing both reciprocal and also uni-directional collateral sensitivities. The effect size of collateral sensitivity, measured as fold-$IC_{75}$, differed substantially among replicates (*Figure 5a*, *Figure 5—figure supplement 1*; *Figure 5—source data 1*). In the current study, we now subjected the total of 38 resistant populations to either strong or strong+constrained increases of the second drug (*Figure 5a*) and assessed treatment outcome by measuring population extinction rates and changes in resistance profiles of the surviving populations (*Supplementary file 1*–Figure 5-supplementary tables 1-2, *Figure 5—figure supplement 1—source data 1*).

Our validation experiment now demonstrated that extinction events were frequent and occurred significantly more often when selection for the original resistance was maintained by the presence of both drugs (extinction in constrained *vs.* strong environments, $\chi^2$ = 16.204, *df* = 1, p<0.0001; *Figure 5b*; *Figure 5—source data 2*). Extinction rates showed pronounced variation by drug identity and order. For example, CIP-resistant populations all survived when challenged with either of the aminoglycosides (STR, GEN), whereas more than 50% of these populations went extinct when exposed to β-lactams (CAR, PIT, *Figure 5b*). We specifically assessed the importance of drug target, collateral sensitivity effect size (fold-$IC_{75}$), and relative growth rate under drug-free conditions, using a generalized linear model (GLM; *Figure 6—source data 1*). We extracted data on the relative exponential growth rate of the resistant populations in drug-free media from our previous publication (*Barbosa et al., 2017*). Our analysis revealed that variation in extinction was significantly associated with only the molecular target of the second antibiotic (GLM, combined extinction, n = 14 drug switches, *F* = 11.016, p=0.011; *Supplementary file 1*–Figure 6–supplementary table 1, *Figure 6—source data 1*), but not with growth rate, extent of collateral sensitivity, or target of the first antibiotic. Extinction frequencies were on average twice as high in the six treatments that switched to β-lactam antibiotics that target the cell wall, as compared to the eight treatments that switched to aminoglycoside antibiotics that target the ribosome (*Figure 6a*). It thus appears that extinctions in

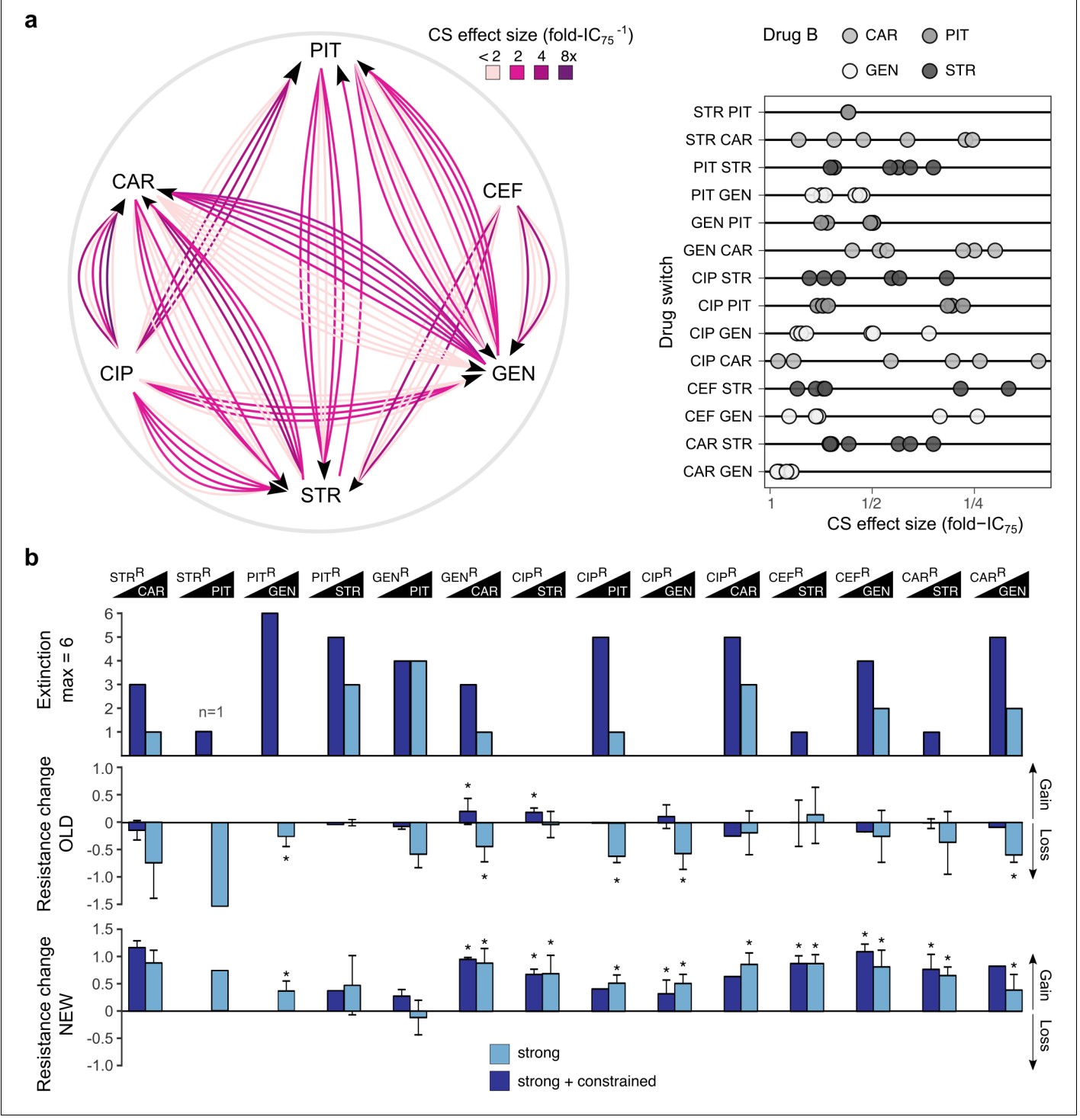

**Figure 5.** Evolutionary stability of collateral sensitivity in a larger set of drug switches. (a) Fourteen collateral sensitivity treatments were tested using 38 distinct resistant populations as starting points. The resistant populations differed with respect to the effect size of collateral sensitivity. The left panel illustrates the treatment directions and effect size variation in different shades of purple, while the right panel shows a quantitative presentation of effect size variation. (b) Total extinction events, and endpoint resistance changes to drug A (here labeled OLD) and B (here labeled NEW) after treatment with drug B using strong concentration increases in the absence (strong) or the presence of drug A (strong + constrained). Mean ±CI95, n = 1–6 biological replicates (differences due to extinction). Asterisks indicate significant changes in resistance (one-sample *t*-test, μ = 0, FDR-adjusted probabilities). CS, collateral sensitivity; CAR, carbenicillin; GEN, gentamicin; PIT, piperacillin with tazobactam; STR, streptomycin; CIP, ciprofloxacin; CEF, cefsulodin; superscript R denotes resistance. The following supplementary figure, tables and source data are available for *Figure 5*: *Figure 5—*

*Figure 5 continued on next page*

*Figure 5 continued*

*figure supplement 1*, *Supplementary file 1*-Figure 5–supplementary tables 1-2, *Figure 5—source data 1*, *Figure 5—source data 2*, *Figure 5—source data 3*, and *Figure 5—figure supplement 1—source data 1*.

The online version of this article includes the following source data and figure supplement(s) for figure 5:

**Source data 1.** Effect size of tested collateral sensitivities.
**Source data 2.** Extinction of experimentally evolved populations.
**Source data 3.** Evolved resistance changes of experimentally evolved populations.
**Figure supplement 1.** Collateral sensitivity.
**Figure supplement 1—source data 1.** Dose-response curves data of surviving populations in the generalized experiment.

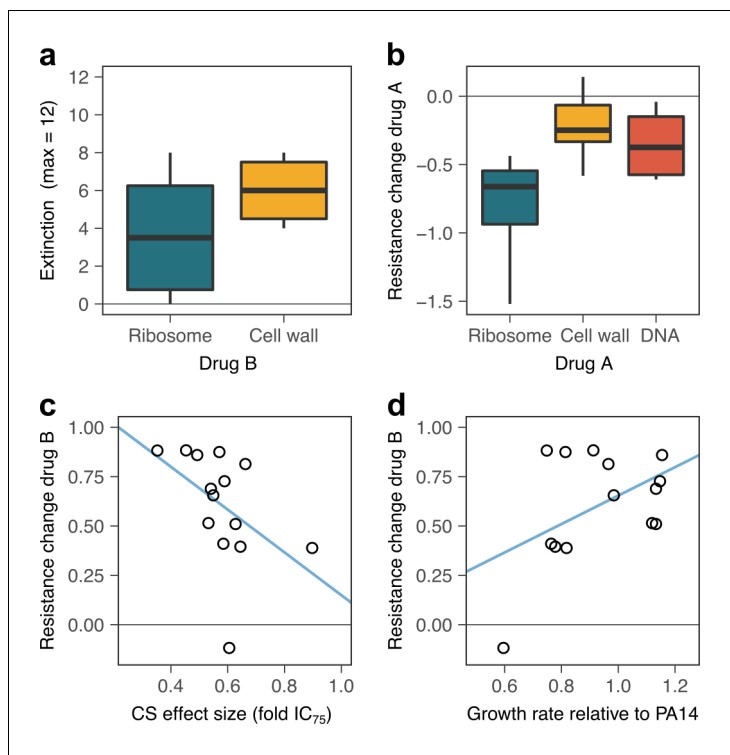

**Figure 6.** Predictors of evolutionary stability of collateral sensitivity in a larger set of drug switches. (**a**) Extinction was more likely when treatment was switched to β-lactam antibiotics that target the cell wall, as compared to aminoglycoside antibiotics that target ribosomes (box plot, n = 6 drug switches for ribosome, n = 8 treatments for the cell wall). Combined extinction events from both treatment groups are reported. (**b**) Surviving, evolved lineages showed stronger re-sensitization when the first antibiotic targeted the ribosome (box plot, n = 4 drug switches starting with ribosome inhibition, n = 4 drug switches starting with cell-wall inhibition, and n = 6 drug switches starting with DNA gyrase inhibition). (**c**) Evolution of new resistance was significantly positively associated with the average effect size of collateral sensitivity, as measured by fold-change of $IC_{75}$. Values of sensitivity increase to the left. (**d**) Average initial growth rate were significantly associated with the resistance gains, with low resistance gains associated with large general adaptation trade-offs. CS, collateral sensitivity. Blue lines in c and d provide an illustration of the linear association between the respective variables. The following supplementary tables and source data are available for (*Figure 6*: *Supplementary file 1*-Figure 6-supplementary tables 1-2, and *Figure 6—source data 1*; *Figure 6—source data 2*).

The online version of this article includes the following source data for figure 6:

**Source data 1.** Summary of total extinction in the generalized experiment in relation to average initial collateral sensitivity effects and average growth rate.
**Source data 2.** Summary of average resistance changes in the generalized experiment in relation to average initial collateral sensitivity effects and average growth rate.

our experiments are stochastic events that are mainly influenced by the target of the current antibiotic, but not the preceding evolutionary conditions. Taken together, we conclude that switching from one to two drugs generally increases treatment potency and that this is influenced by drug order and identity. The preferred drug order is to use a β-lactam as the second antibiotic, which is consistent with the detailed dynamics on the clonal level.

Resistance gains to drug B were frequently associated with re-sensitization to the original drug A in the unconstrained treatments, while resistance to the new drug B increased in all 14 tested cases (leading to changes of 2 up to >64 times the $IC_{90}$ of the starting populations; *Supplementary file 1*– Figure 5–supplementary tables 1 and 2). In detail, we observed significant antibiotic re-sensitization for 5 of the 14 cases (36%, *Figure 5b*, *Figure 5—source data 3*, *Supplementary file 1*–Figure 5– supplementary table 1, *Figure 5—source data 1*; Note that the single available replicate for STR>PIT showed a similar trend but could not be evaluated statistically). Within reciprocal collateral sensitivity, re-sensitization was not always observed in both directions, as illustrated by CAR/STR and STR/PIT. Re-sensitization did not occur in the constrained treatments. We noticed some differences between evolutionary outcomes of drug switches in clonal and population experiments. The resistance changes showed an overall categorical agreement of 75% (9/12 comparisons; no comparisons for STR>PIT, as this switch was tested with a single replicate). Yet, clonal evolution produced re-sensitization for only one sequential treatment direction for CAR/GEN, whereas population evolution showed re-sensitization for both directions. That re-sensitization occurred in both directions at the population level is expected if there was a residual frequency of cells lacking the collateral-sensitivity mutation in the inoculum. The population data further highlighted a lack of re-sensitization for PIT>STR, where we previously observed re-sensitization on the clonal level, albeit not at both concentration regimes. In addition, there is a mild significant increase of resistance to GEN in the strong constrained treatments in GEN>CAR, compared to no change. Overall our results indicate that re-sensitization is frequent for populations with collateral sensitivity, but not imperative (*Figure 5b*).

What are the drivers of the observed resistance gains and re-sensitization? Our statistical analysis revealed that the extent of re-sensitization in the unconstrained treatments was significantly associated to equal degrees with the target of drug A (GLM, resistance against drug A, n = 14, F = 15.27, p=0.0019; *Supplementary file 1*–Figure 6–supplementary table 2, *Figure 6—source data 2*), the effect size of collateral sensitivity (F = 15.91, p=0.004), and growth rate in drug-free media (F = 18.38, p=0.0027), while target of drug B did not have a significant influence. Re-sensitization was stronger in drug switches that started with an aminoglycoside, as compared to β-lactam, and fluoroquinolone (*Figure 6b*). This result is consistent with the drug-dependent negative epistasis (*Figure 4*), and validates the relative stability of collateral sensitivity switches from aminoglycosides to β-lactams. Growth rate in drug-free conditions and collateral sensitivity effect size showed small negative effects, indicating that they decelerate re-sensitization. In contrast, the extent of resistance gains in unconstrained treatments was not associated with drug targets of either drug A or B, but instead significantly associated with both the initial effect size of collateral sensitivity (GLM, resistance against drug B, n = 14, F = 6.71, p=0.0321; *Supplementary file 1*-Figure 6–supplementary table 2, *Figure 6—source data 2*), and the initial growth rate of the resistant population in drug-free media (GLM, resistance against drug B, n = 14, F = 7.90, p=0.0228). Collateral sensitivity effect size showed a negative association with the resistance gains to drug B (*Figure 6c*). High sensitivity thus appears to be prone to more rapid resistance gains, possibly because it opens a fitness space for rapid adaptation. The population resistance changes against drug B were generally larger than the initial collateral sensitivity (*Figure 5b*; *Figure 5—source data 3*); *Supplementary file 1*-Figure 5-supplementary tables 1-2; evolved lineages showed higher resistance than the PA14 wild type), indicating that they were not solely due to competition of pre-existing sub-populations, but involved new genetic adaptations. As observed above, the presence of a general adaptation trade-off (measured as reduction in growth rate relative to ancestral PA14 in drug-free environments) inhibited rapid adaptation (*Figure 6d*). Taken together, our results at the population level suggest that the stability of collateral sensitivity switching is shaped predominantly by drug targets, but additionally influenced by the effect size of collateral sensitivity and a general adaptation trade-off. The most stable collateral sensitivity was identified for the switch from aminoglycoside to β-lactam. Within a particular drug switch, the highest stability was achieved for low collateral sensitivity effect sizes. Large adaptation trade-offs tended to inhibit evolutionary change, that is both desirable re-sensitization, and new resistance. The conclusions are consistent with the data obtained from the clonal evolution experiments.

# Discussion

Collateral sensitivity is a pervasive feature of resistance evolution, but its potential for medical application is currently debated (*Nichol et al., 2019*; *Podnecky et al., 2018*; *Roemhild and Schulenburg, 2019*). Its promise as a treatment focus is dependent on the premise that the exploited trade-off is evolutionarily stable and cannot be easily overcome. As a consequence, it should either drive bacterial populations to extinction or minimize the emergence of multidrug resistance by re-sensitization to one of the antibiotics. We here tested the validity of these key predictions with the help of evolution experiments and the model pathogen *P. aeruginosa*. We found that the effective exploitation of evolved collateral sensitivity in sequential therapy is contingent on drug order and combination, collateral sensitivity effect size, general adaptation trade-offs, and also epistatic genetic interactions.

Evolved reciprocal collateral sensitivity generally limited bacterial adaptation. Adaptation was particularly constrained in treatments that switched to a β-lactam, as reflected by the elevated extinction rates. The effect was strongest when the first antibiotic was maintained and a second was added. This finding may point to a promising, yet currently unexplored treatment strategy, namely single-drug therapy followed by combination therapy (for example the addition of a β-lactam), that can maximize exploitation of the evolutionary trade-off. Yet, extinction rates were still high under unconstrained conditions, when drugs were replaced, and in spite of a relatively mild selection intensity. In the detailed analysis with the CAR/GEN drug pair, we observed higher extinction and slower growth improvements in strong, compared to mild dose increases. This finding is generally consistent with previous studies, performed in different context, in which narrowed mutation space upon fast environmental deterioration increased extinction frequencies (*Bell and Gonzalez, 2011*; *Lindsey et al., 2013*). Interestingly, extinction rates are often not reported as an evolutionary outcome in related studies, possibly because of a different main focus of the study (*Yen and Papin, 2017*), or because extinction could not be recorded due to the particular experimental set-up (i.e., usage of a morbidostat; *Yoshida et al., 2017*). Considering that antimicrobial therapy usually aims at elimination of bacterial pathogens and extinction frequencies are known from previous evolution experiments to vary among treatment types (*Barbosa et al., 2018*; *Hansen et al., 2017*; *Roemhild et al., 2018*; *Torella et al., 2010*), their consideration should help us to refine our understanding of treatment efficacy.

In the treatments that replaced drugs, evolutionary stability of the resistance trade-off was determined by drug order. In both datasets (clonal and population-level evolution), we observed high relative stability for collateral sensitivity treatments switching from an aminoglycoside to a β-lactam. Our detailed analysis for the CAR/GEN pair at the clonal level demonstrates that this stability may be caused by slower adaptation (*Figure 3a*), and efficient re-sensitization (*Figure 3c*). Also, the slow adaptation rates may be the result of selection favoring the reversal of two evolutionary trade-offs (*Figures 3e* and *6d*) together with a relative paucity of accessible resistance mutations. The latter is highlighted by several instances of re-mutation of *pmrB*, where the effect of the ancestral collateral-sensitivity SNP was countered by additional larger deletions, potentially leading to a loss of function of the same gene (*Figure 4b*). The efficient re-sensitization that was observed during switches from an aminoglycoside to a β-lactam may be explained by pleiotropy and drug-specific negative epistasis (*Figure 4c and d*). On the other hand, collateral sensitivity treatments that switch to aminoglycosides, tended to show lower stability, as reflected by the lower levels of extinction (*Figure 6a*) and the lack of re-sensitization (*Figure 6b*) at the population level. For instance, the collateral sensitivity treatments CAR>STR and CIP>STR were frequently countered by the evolving bacteria. These treatments hardly resulted in extinction but allowed for rapid loss of susceptibility to both drugs (*Figure 5b*). While our data cannot fully explain the high instability, it appears that small values for a general adaptation trade-off (i.e., indicated by reduced growth rates in drug-free environments), which are also frequently observed for CIP resistance mutations (*Huseby et al., 2017*), accelerated adaptation (*Figure 6d*). Overall, our finding of strong variation in the stability of collateral sensitivity treatments in pathogenic *P. aeruginosa* indicate the importance for a careful evaluation of new treatment options.

Our detailed analyses for GEN>CAR emphasize the importance of epistasis for the stability of collateral sensitivity. A recent publication confirmed that the expression of particular collateral sensitivity mutations strongly depended on the genetic background and could even cause opposite effects in the closely related species of *E. coli* and *Salmonella enterica* due to epistasis (*Apjok et al., 2019*).

Our work on the GEN>CAR switch now shows, that epistasis is of high importance also for the temporal stability of antibiotic sensitivity. Here, drug re-sensitization in the unconstrained treatments was likely dependent on negative epistasis among pleiotropic resistance mutations. Mutations in *pmrB* and the efflux regulators *nalC* and *nalD* interacted negatively with each other and caused a complete re-sensitization of bacteria that were previously resistant against GEN. While re-sensitization reliably occurred for the GEN>CAR treatment, it did not occur in the reverse case. Similar examples of antibiotic re-sensitization were previously reported for *E. coli* and *P. aeruginosa*, but these relied on different mechanisms. For *E. coli*, repeated alternation between two antibiotics led to re-sensitization as a consequence of clonal interference between variants in two genes, *secD* and/or *basB* (*Yoshida et al., 2017*). The change between drugs prevented fixation of the competing variants, thus maintaining pleiotropic alleles and thereby the allele causing resistance to one drug and hypersensitivity to the other (*Yoshida et al., 2017*). In the previous example for *P. aeruginosa*, hypersensitivity to a β-lactam depended on an expression imbalance of the MexAB-OprM and the MexEF-OprN efflux systems after exposure to a fluoroquinolone (*Maseda et al., 2004*; *Sobel et al., 2005*; *Yen and Papin, 2017*). Interestingly, partial re-sensitization against the aminoglycoside tobramycin was dependent on inducible resistance, a phenomenon mediated by the MexXY-OprM efflux pump, whereby expression, and consequently resistance, is induced by the presence of the drug, but then reverted after its removal (*Hocquet et al., 2003*; *Yen and Papin, 2017*). We conclude that our finding of negative epistasis between pleiotropic resistance mutations is a previously unknown mechanism underlying re-sensitization. Whilst positive epistasis can substantially amplify resistance gains (*Wistrand-Yuen et al., 2018*), negative epistasis can limit evolutionary trajectories (*Weinreich et al., 2006*), thus possibly contributing to efficacy of treatment in our case.

The mutations observed in this study are commonly associated with variants observed in clinical isolates, particularly in those obtained from cystic fibrosis patients (*Hancock and Speert, 2000*; *Jansen et al., 2016*; *Marvig et al., 2015*; *Tueffers et al., 2019*). Both efflux regulators (including *nalC* and *nalD*) and two-component regulatory systems (mainly *pmrAB* and *phoQF*) were repeatedly reported to be associated with intermediate and highly resistant isolates of *P. aeruginosa, E. coli, Acinetobacter baumannii* and other pathogenic species against colistin and aminoglycosides (*Cao et al., 2004*; *Gerson et al., 2019*; *Sato et al., 2018*). This overlap suggests that the negative epistasis between the genes involved in resistance against β-lactam and aminoglycosides observed here could also be encountered and exploited in clinical settings.

We anticipate that the findings of our study could help to guide the design of sustainable antibiotic therapy that controls the infection, whilst reducing the emergence of multidrug resistance. In principle, the refined exploitation of collateral sensitivity could represent a promising addition to new evolution-informed treatment strategies, including as alternatives specific combination treatments (*Barbosa et al., 2018*; *Chait et al., 2007*; *Rodriguez de Evgrafov et al., 2015*; *Gonzales et al., 2015*; *Munck et al., 2014*), fast sequential therapy (*Nichol et al., 2015*; *Yoshida et al., 2017*), or treatments utilizing negative hysteresis (*Roemhild et al., 2018*). The success of this treatment strategy depends on several key factors. One critical prerequisite is that collateral sensitivity does evolve in response to treatment, which may vary depending on alternative evolutionary paths to resistance against the initially used drug A (*Barbosa et al., 2017*; *Nichol et al., 2019*). Our new data additionally suggest that treatment can be further optimized by switching from an aminoglycoside to a β-lactam and/or by focusing on drugs, for which resistance is associated with large general adaptation trade-offs, and/or by using drug combinations with small collateral sensitivity effect sizes. These new insights clearly warrant further evaluation, for example by using clinical isolates of *P. aeruginosa* and/or patient-related test conditions.

## Materials and methods

### Material

All experiments were performed with *P. aeruginosa* UCBPP-PA14 (*Rahme et al., 1995*) and clones obtained from four antibiotic-resistant populations: *CAR-10, GEN-4, PIT-1* and *STR-2* (*Barbosa et al., 2017*). The resistant populations were previously selected for high levels of resistance against protein synthesis inhibitors from the aminoglycoside family, gentamicin (GEN; Carl Roth, Germany; Ref. HN09.1) and streptomycin (STR; Sigma-Aldrich, USA; Ref. S6501-5G), or

alternatively cell-wall synthesis inhibitors from the β-lactam family, carbenicillin (CAR; Carl Roth, Germany; Ref. 6344.2) and piperacillin/tazobactam (PIT; Sigma-Aldrich, USA; Refs. P8396-1G and T2820-10MG). Resistant clones were isolated by streaking the resistant populations on LB agar plates supplemented with antibiotics and picking single colonies after an overnight growth at 37°C. Antibiotic stocks were prepared according to manufacturer instructions and frozen in aliquots for single use. Evolution experiments and resistance measurements were performed in liquid M9 minimal media supplemented with glucose (2 g/l), citrate (0.5 g/l) and casamino acids (1 g/l).

## Measurements of reciprocal collateral sensitivity

The previously reported collateral sensitivity trade-off (Barbosa et al., 2017) was confirmed for this study, by measuring sensitivity of the resistant populations *CAR-10* to GEN, *GEN-4 to CAR*, *PIT-1* to STR, and *STR-2* to PIT, in comparison to PA14. Populations were grown to exponential phase, standardized by optical density at 600 nm ($OD_{600} = 0.08$), and inoculated into 96-well plates (100 µl volumes, $5 \times 10^6$ CFU/ml) containing linear concentrations of antibiotics (10 concentrations, eight replicates each). Antibiotic concentrations were spatially randomized. Plates were incubated at 37 °C for 12 hr, after which growth was measured by $OD_{600}$ with a BioTek plate reader. Antibiotic susceptibility was quantified from dose-response curves using the lowest concentration required to inhibit growth by a defined value compared to wild type growth in drug free medium. $IC_{95}$ (inhibitory concentration 95) refers to the smallest concentration required to inhibit growth by 95%. The metrics $IC_{90}$ and $IC_{75}$ refer to the concentrations that inhibit growth by 90% or 75%, respectively. Inhibitory concentrations were determined from the dose-response data using linear interpolation between the two closest $OD_{600}$ values, as inferred with the *approx* function in the statistical environment *R*. In our case, the $IC_{75}$ or $IC_{90}$ metrics show higher accuracy and precision than the commonly used metric of the minimal inhibitory concentration (MIC, equivalent to $IC_{100}$), because we inferred growth characteristics from $OD_{600}$ measurements, which are subject to unfavorable signal-to-noise ratios close to OD values of zero and thus close to the $IC_{100}$ condition. Please note that analysis of IC75 and IC90 values produced consistent results (e.g., *Supplementary file 1*-Figure 6-supplementary tables 1 and 2).

## Experimental evolution initiated with resistant clones

To test the evolutionary stability of reciprocal collateral sensitivity, we challenged clones from previously evolved resistant populations with increasing concentrations of new antibiotics against which the resistant populations showed hypersensitivity (so called collateral sensitivity): *CAR-10* with GEN, *GEN-4* with CAR, *PIT-1* with STR, and *STR-2* with PIT. Stability was assessed with 12 day evolution experiments using a serial transfer protocol (100 µl batch cultures, 2% serial transfers every 12 hr; the starting population size for the different populations was approx. $10^6$ CFU/ml), as previously described (Roemhild et al., 2018). Each population was evaluated with eight replicate populations (4 clones x two technical replicates distributed in two plates: plate A and plate B) for each of 5 treatment groups: (i) untreated controls; linearly increasing concentration of hypersensitive antibiotic to a low level (ii) or high level (iii), without maintaining selection for previous resistance (unconstrained evolution); or linearly increasing concentration of hypersensitive antibiotic to a low level (iv) or high level (v), with simultaneous selection for previous resistance (constrained evolution). Concentration increases were started with defined initial inhibition levels of 50% ($IC_{50}$) and concluded when concentrations were above the $IC_{95}$ of the hypersensitive strain (mild increases) or $IC_{95}$ of the wild type PA14 strain (strong increases), as specified in *Supplementary file 1*-Supplementary Table 1. Antibiotic selection was applied in 96-well plates and population growth was monitored throughout treatment by continuous measurements of $OD_{600}$ in 15 min intervals (BioTek Instruments, USA; Ref. EON; 37 °C, 180 rpm double-orbital shaking). Extinction frequencies were determined at the end of the experiment by counting cases in which no growth was observed after an additional transfer to antibiotic-free media and 24 hr of incubation. Surviving evolved populations were frozen at −80 °C in 10% (v/v) DMSO, at the end of the experiment.

## Relative biomass

The continuous measurements of optical density during treatment provided a detailed growth trajectory that accurately describes the dynamics of resistance emergence. Relative biomass was defined

as total optical growth relative to untreated control treatments, and was calculated by the ratio of the areas under the time-OD curves of treated compared to untreated controls that are passaged in parallel, as previously described (*Roemhild et al., 2018*).

## Resistance of evolved populations

Resistance of evolved populations was measured for the respective antibiotic pairs (GEN/CAR or STR/PIT), as described above, but using two-fold concentrations (1/4 to 8x the MIC of the starting clone). The respective starting clones of each evolved population served as controls and were measured in parallel. Resistance changes were quantified by subtracting the area under the dose-response curve of the evolved populations from that of the ancestral clones. Positive values indicate that the evolved lineages are more resistant than their ancestor, values close to zero indicate equivalent resistance levels, and negative values denote a loss of resistance. The cases of re-sensitization against GEN were validated by repeating the measurements, whereby the PA14 ancestor was included as an additional control (*Figure 3—figure supplement 2*).

## Growth rate analyses

Maximum exponential growth rates of evolved and ancestral populations were calculated from growth curves in drug-free media, using a sliding window approach. For measurements, sample cultures were diluted 50x from early stationary phase into 96-well plates (100 µl total volume) and growth was measured by $OD_{600}$ every 15 min for 12 hr. Growth rate were calculated from log-transformed OD data for sliding windows of 1 hr, yielding two-peaked curves indicating initial growth on glucose and citrate. The reported values the maximum values of the first, larger peak. The values reported in *Figure 3* are the changes of growth rate in evolved populations relative to their resistant ancestors. These values are taken as a proxy for a general adaptation trade-off, which is distinct to the collateral sensitivity trade-off.

## Genomics

We re-sequenced whole genomes of 5 starting clones (CAR-10 clone 2, GEN-2 clones 1–4), and 21 evolved populations (all descendants of these clones from plate B, including five untreated evolved control populations and 16 populations adapted to different treatment conditions) using samples from the end of the evolution experiments. Frozen material was thawed and grown in 10 ml of M9 minimal medium for 16–20 hr at 37 °C with constant shaking. Genomic DNA was extracted using a modified CTAB buffer protocol (*von der Schulenburg et al., 2001*) and sequenced at the Institute for Clinical Microbiology, Kiel University Hospital, using Illumina HiSeq paired-end technology with an insert size of 150 bp and 300x coverage. For the genomic analysis, we followed an established pipeline (*Jansen et al., 2015*). Briefly, reads were trimmed with Trimmomatic (*Bolger et al., 2014*), and mapped to the UCBPP-PA14 reference genome (available at http://pseudomonas.com/strain/download) using bwa and samtools (*Li and Durbin, 2010*; *Li et al., 2009*). We used MarkDuplicates in Picardtools to remove duplicated regions for single nucleotide polymorphisms (SNPs) and structural variants (SVs). To call SNPs and small SV we employed both heuristic and frequentist methods, only for variants above a threshold frequency of 0.1 and base quality above 20, using respectively VarScan and SNVer (*Wei et al., 2011*). Larger SVs were detected by Pindel and CNVnator (*Abyzov et al., 2011*; *Ye et al., 2009*; *Ye et al., 2009*). Variants were annotated using snpEFF (*Cingolani et al., 2012*), DAVID, and the *Pseudomonas* database (http://pseudomonas.com). Variants detected in the untreated evolved populations were removed from all other populations and analyses as these likely reflect adaptation to the lab media and not treatment. The fasta files of all sequenced populations here are available from NCBI under the BioProject number: PRJNA524114.

## Genetic manipulation

To understand re-sensitization, we analyzed candidate mutations from the GEN > CAR switch. The *nalD* mutation 1551588G > T (resulting in amino acid change p.T11N, as observed in replicate populations b24_G8, b24_D9, and b24_A9) was introduced into the PA14 genetic background using a scar-free recombination method (*Trebosc et al., 2016*). The same techniques were previously used to construct the mutants *nalC* (deletion 1391016–1391574) and *pmrB* (5637090T > A, resulting in amino acid change p.V136E) in the PA14 ancestor background (*Barbosa et al., 2017*). Based on

these mutants and with the same techniques, we constructed the double mutants *pmrB, nalD* (pmrB p.V136E + nalD p.T11N), and *pmrB, nalC* (pmrB p.V136E + nalC deletion c.49–249, as observed in population b24_F7). Genetic manipulation and confirmation by sequencing was performed by Bio-Versys AG, Hochbergerstrasse 60 c, CH-4057 Basel, Switzerland.

### Epistasis analysis

Resistance of constructed mutant strains was measured in direct comparison to wildtype PA14, as described above. Relative fold-changes in MIC were calculated from dose-response curves. The expected relative resistance of the double mutants was calculated by multiplication of the mutation's individual effects, as previously described (*Wong, 2017*). For example, if mutation A conferred a 2-fold increase in resistance and mutation B conferred a 4-fold increase of resistance, the expected resistance of the double mutant AB would be $2 \times 4 = 8$. A deviation from this null model indicates epistasis, which can be either positive (greater resistance than expected) or negative (lesser resistance than expected).

### Validation of main findings through a repetition of evolution experiments using resistant populations as starting material

The evolutionary stability of collateral sensitivity in genetically diverse populations was investigated by using the same general procedure as described above (section 'Experimental evolution initiated with resistant clones'), but using an inoculum of roughly $10^7$ cells instead of a single clone. This experiment was reduced to the treatments groups 'strong' and 'strong+constrained', and performed for a total of 38 ($6 \times 6 + 2$) different resistant starting populations from our previous publication (*Barbosa et al., 2017*). Six replicate populations each from previous evolution for resistance to CAR, GEN, STR, PIT, ciprofloxacin (CIP) and cefsulodin (CEF), were challenged with increasing concentrations of antibiotic against which they showed collateral sensitivity. The population were each evolved against two new antibiotics. Due to variation in collateral sensitivity profiles among replicate populations, a seventh population had to be selected for CAR and PIT to assemble a set of 6 collaterally-sensitive populations for each of the switching directions. Only one STR-resistant population showed collateral sensitivity to PIT so that this test was conducted with a single replicate. CIP-resistant populations showed general collateral sensitivity and were tested against four new antibiotics. In total, we thus arrived at 79 ($14 \times 6 - 5$) evolutionary switches to collateral sensitivity.

### Statistical analysis for association with evolutionary stability

To test for association of predictive factors with evolutionary stability, we used a generalized linear model (GLM) analysis, because it allows us to combine an evaluation of both categorical and continuous predictors and to assess the influence of each factor in consideration of the contributions of the other factors (which is not possible when using for example correlation analysis). For our analysis, we used the functions *lm* and *anova* in the statistical environment *R* and the main effects model: response ~target drug A + target drug B + collateral sensitivity effect size + drug-free relative growth rate.

## Acknowledgements

We thank D I Andersson, R Kishony, C Kost, and V Lázár for feedback on the manuscript. Genome sequencing was performed by G Hemmrich-Stanisak and M Vollstedt from the Institute of Clinical Molecular Biology in Kiel, as supported by the DFG Cluster of Excellence EXC 306 'Inflammation at Interfaces'. This research was funded by the Deutsche Forschungsgemeinschaft (DFG, German Research Foundation) individual grant SCHU 1415/12 (to HS) and also under Germany's Excellence Strategy – EXC 22167–39088401 (Excellence Cluster Precision Medicine in chronic Inflammation; HS, PR), the Leibniz Science Campus Evolutionary Medicine of the Lung (EvoLUNG; HS, CB), the International Max-Planck-Research School for Evolutionary Biology (CB, RR), and the Max-Planck Society (HS, RR).

## Additional information

### Funding

| Funder | Grant reference number | Author |
|---|---|---|
| Deutsche Forschungsgemeinschaft | SCHU 1415/12-1 | Hinrich Schulenburg |
| Deutsche Forschungsgemeinschaft | EXC 22167-39088401 | Philip Rosenstiel<br>Hinrich Schulenburg |
| Leibniz-Gemeinschaft | EvoLUNG | Camilo Barbosa<br>Hinrich Schulenburg |
| Max-Planck-Gesellschaft | IMPRS Evolutionary Biology | Camilo Barbosa<br>Roderich Römhild |
| Max-Planck-Gesellschaft | Fellowship | Hinrich Schulenburg |

The funders had no role in study design, data collection and interpretation, or the decision to submit the work for publication.

### Author contributions

Camilo Barbosa, Roderich Römhild, Conceptualization, Investigation, Visualization, Writing—original draft, Writing—review and editing; Philip Rosenstiel, Investigation, Methodology; Hinrich Schulenburg, Conceptualization, Supervision, Funding acquisition, Writing—original draft, Writing—review and editing

### Author ORCIDs

Camilo Barbosa (iD) https://orcid.org/0000-0003-0433-9311
Roderich Römhild (iD) https://orcid.org/0000-0001-9480-5261
Philip Rosenstiel (iD) https://orcid.org/0000-0002-9692-8828
Hinrich Schulenburg (iD) https://orcid.org/0000-0002-1413-913X

### Decision letter and Author response

Decision letter https://doi.org/10.7554/eLife.51481.sa1
Author response https://doi.org/10.7554/eLife.51481.sa2

## Additional files

### Supplementary files

• Supplementary file 1. File with all supplementary tables.

• Transparent reporting form

### Data availability

Sequencing data have been deposited at NCBI under the BioProject number PRJNA524114. All other data generated or analysed during this study are included in the manuscript and supporting files. Source data files have been provided for all figures.

The following dataset was generated:

| Author(s) | Year | Dataset title | Dataset URL | Database and Identifier |
|---|---|---|---|---|
| Barbosa C, Roemhild R, Rosenstiel P, Schulenburg H | 2019 | Evolutionary stability of antibiotic collateral sensitivity | https://www.ncbi.nlm.nih.gov/bioproject/?term=PRJNA524114 | NCBI Bioproject, PRJNA524114 |

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
