## [Decision Letter]

**Acceptance summary:**

Despite this significant progress in the study of collateral sensitivity over the recent years, there are several key questions about its clinical utility. This interesting work found that exploitation of evolved collateral sensitivity in sequential therapy is contingent on drug order, epistatic genetic interactions and consequent evolutionary trade-offs. These findings could contribute to the design of new antibiotic therapies with the aim of limiting the emergence of multidrug bacteria.

**Decision letter after peer review:**

[Editors’ note: a previous version of this study was rejected after peer review, but the authors submitted for reconsideration. The first decision letter after peer review is shown below.]

Thank you for submitting your work entitled "Evolutionary stability of collateral sensitivity to antibiotics in the model pathogen *Pseudomonas aeruginosa*" for consideration by *eLife*. Your article has been reviewed by a Senior Editor, a Reviewing Editor, and three reviewers. The following individuals involved in review of your submission have agreed to reveal their identity: Pal Johnsen (Reviewer #1); Balint Csorgo (Reviewer #2); Craig MacLean (Reviewer #3).

Our decision has been reached after consultation between the reviewers. Based on these discussions and the individual reviews below, we regret to inform you that your work will not be considered further for publication in *eLife*. As you can see from the reports, the reviewers found the topic of potential interest, but they raised serious concerns about the generality of the conclusions reached and clinical implications thereof. The reviewers noted that the scope of the study is rather limited, for two main reasons. Only a single source population for each selection treatment and only two collateral-sensitive drug pairs were studied. However, we would be willing to look at a substantially revised version should the authors decide to complete all the experiments suggested by the reviewers.

Reviewer #1:

This is a well written and very interesting manuscript addressing evolutionary stability of reciprocal collateral sensitivity (CS) in *Pseudomonas aeruginosa*. This is an important and currently un-explored aspect of CS networks and absolutely critical for potential future clinical application as a strategy to offset antibiotic resistance evolution.

The authors demonstrate that evolutionary stability of reciprocal CS phenotypes depend on not only pairs of drugs but also the order of administration during experimental evolution. They conclude that this is due to fitness cost variations and epistatic interactions between pleiotropic resistance mutations.

Essential revisions:

1) The authors conclude that for the first pair of drugs investigated (STR-PIT) frequent extinctions (51% of the populations) during evolution was due to strong genetic constraints against an evolutionary response to overcome CS. Beyond the initial phenotype and number of extinct populations I find this claim largely unsubstantiated. In full awareness of how difficult it would be to include a "non-CS control experiment" it would in my opinion benefit the paper to know what actually happened in the extinct versus non-extinct populations. Could not alternative hypotheses be that reduced susceptibility (STR-PIT) require accumulation of more mutations than for the CAR-GEN pair or if classical synergy between STR and PIT for the combination exposures led to more extinctions (aminoglycosides and β-lactams are often synergetic, including in *Pseudomonas*)?

2) The authors demonstrate elegantly an example of negative epistasis between pleiotropic resistance mutations and this is an important contribution to our mechanistic understanding of CS/hypersensitivity where very little mechanistic details exist. However, since new treatment strategies are frequently mentioned throughout the manuscript, I believe the manuscript would benefit from a link to clinical reality. Do the observed mutations also emerge during therapy or are they specific for the experimental conditions? How does the observed resistance levels relate to existing clinical breakpoints?

3) The authors argue, and partially conclude, that if the costs of resistance are high, it is more difficult to overcome CS. If I understand correctly, this is based on Figures 3E,F and Supplementary Figure 2, where relative growth rates are used as a proxy for relative fitness before and after evolution. First, these experiments are done in antibiotic free media and not in conditions experienced during evolution. Second, would not the proper comparisons be with the PIT-STR pairs, by analyzing data from the evolved populations (that did not go extinct) as well as doing relative fitness measurements with the pairs of genetically re-constructed mutants to avoid bias from environmental adaptation?

Reviewer #2:

In their work "Evolutionary stability of collateral sensitivity to antibiotics in the model pathogen *Pseudomonas aeruginosa*", Barbosa and colleagues explore the effects of adapting antibiotic resistant bacteria to other antibiotics against which they had previously developed hypersensitivity during their original adaptation. As the emergence of antibiotic resistant bacteria threatens to become an increasingly severe health crisis, multiple strategies are needed to combat this. One such strategy is the exploitation of collateral-sensitivity phenotypes of drug resistant bacteria, an approach which has picked up interest over the last 5 years or so. This work aims to study the long-term stability of these interactions, an aspect about which little is known about, therefore I think the general aim is timely and relevant.

Overall, I think the experiments are designed well, the paper is well written, and the general conclusions are supported by the experimental evidence presented. The authors found that attempting to exploit collateral sensitivity phenotypes through sequential antibiotic treatments had greatly varying effects, depending on which antibiotic was used, in what order antibiotic pairs were used, and whether combination therapy was employed in the second treatment or not. The PIT/STR antibiotic pair lead to high extinction rates while the CAR/GEN pair was not as effective, frequently resulting in the evolution of new resistance mechanisms that allowed either multidrug resistance, or less frequently, resensitization to the original drug. The likely molecular mechanisms responsible for these phenotypes were revealed through whole-genome sequencing. I appreciated their efforts in chromosomally reconstructing a small set of these mutations, which revealed an interesting case of negative epistasis between two resistance mutations that was responsible for the resensitization phenotype.

My questions are the following:

1) I found the scope of the work somewhat limited, as only two collateral-sensitive drug pairs were studied. It would be nice to see the same experimental setup on a greater number of reciprocal, collaterally-sensitive antibiotic pairs to get a better idea of the frequency of pairs that more frequently cause extinction and pairs that lead to resistance to the second antibiotic. From their previous work (Barbosa et al., 2017), this may be e.g. DOR/STR, CAR/PIT, CAR/STR; did the authors examine these strains?

2) On that note, it would have also been interesting to look at the resistance mechanisms of the survivors of the STR/PIT sequential treatments. The authors point out that this analysis was not possible because of the limited number of survivor populations that grew, however by my count, 31 out of 64 populations actually did survive from the various treatment regiments. It would be interesting to see if these populations are multidrug resistant or perhaps cases of resensitization can be observed here as well.

3) I found it interesting that the PIT/STR combination that seems to be the most effective in this study did not show reciprocal collateral sensitivity in the majority of cases in the authors' previous work (Barbosa et al., 2017). Only one out of 10 STR adapted lines was hypersensitive to PIT while cross-resistance arose in 5 of the cases. Can the fact that cross-resistance seems to be more common with this pair in the STR -> PIT direction mean that this combination would actually not be a prudent choice for treatment against *P. aeruginosa* PA14?

Reviewer #3:

In this manuscript, the authors use experimental evolution to investigate the consequences of collateral sensitivities that evolved when *Pseudomonas aeruginosa* evolved resistance to antibiotics in a previously published selection experiment. This paper investigates a hot topic in evolutionary biology (antibiotic resistance) using previously established methods, and this paper clearly makes a novel contribution to our understanding of trade-offs between resistance to alternative antibiotics (collateral sensitivity). The genome sequencing and mutant reconstruction provide good support for the experimental evolution component of the paper, and the use of this approach adds an interesting layer of detail to the paper. However, I have some reservations about the experimental design and data analysis for this paper, and I question the breadth of conclusions that can be drawn from this study.

1) Experimental design and possible pseudo-replication:

The authors setup their selection experiment with 2 replicate cultures of 4 clones from 4 different populations that were previously selected for resistance to CAR, GEN, STR and PIP. I have two criticisms of this design.

a) In their analysis of extinction rates and evolutionary dynamics (Figure 2 and Figure 3), the authors treat the 2 replicates of each clone as being independent of each other, even though they were derived from the same clone. It is difficult to evaluate the impact of this pseudo-replication on the results of the analysis, and this is an issue that the authors need to address.

b) By choosing multiple clones from a single source population, the authors limit the generality of their findings. For example, I do not think that it is a fair conclusion to say that PIT adapted populations tend to go extinct when they are challenged with STR (and vice versa), because all of the PIT adapted clones came from a single population that evolved PIT resistance.

c) Overall, I think that the study would have been stronger if the authors had instead started their selection experiments with 8 independently evolved clones, each from a different source population.

2) Measurements in resistance:

Clinical microbiologists measure antibiotic resistance using standardized techniques to estimate MIC. The authors measure changes in the resistance of evolved populations by measuring changes in the area under the inhibition curve (Figure 3). This data provides a more subtle and quantitative measure of resistance, but this metric lacks a unit and the authors should also measure changes in MIC so that their results can be placed more easily in the context of antibiotic resistance. For example, it is unclear to what extent adaptation to new antibiotics changed resistance to old antibiotics. This data will help clinicians to evaluate the extent to which evolutionary changes in resistance are likely to lead to different outcomes in the clinic.

3) Applicability of findings:

The authors conclude that the findings of their study could help to guide the design of sustainable antibiotic therapies in the clinic (Discussion section). I am not convinced that this is the case, and I think that the authors should be careful about overstating the applied significance of their work. Because the authors used only a single source population for each selection treatment, they cannot make any general conclusions about how different antibiotics collateral sensitivity (i.e. this study does not support the idea that PIT followed by STR will be a generally effective strategy for treating *Pseudomonas* infections). A further limitation to the applicability of this study comes from the fact that both collateral sensitivity combinations relied on using aminoglycoside antibiotics (STR and GEN); these antibiotics are rarely used to treat *Pseudomonas* infections (i.e. https://www.thoracic.org/statements/resources/tb-opi/hap-vap-guidelines-2016.pdf). I think that it is important to be careful about how researchers in this area interpret their findings in the context of clinical use of antibiotics.

[Editors’ note: what now follows is the decision letter after the authors submitted for further consideration.]

Thank you for resubmitting your work entitled "Evolutionary stability of collateral sensitivity to antibiotics in the model pathogen *Pseudomonas aeruginosa*" for further consideration at *eLife*. Your revised article has been favorably evaluated by Patricia Wittkopp (Senior Editor), a Reviewing Editor, and three reviewers. The following individuals involved in review of your submission have agreed to reveal their identity: Balint Csorgo (Reviewer #1); Craig MacLean (Reviewer #2); Pal Johnsen (Reviewer #3).

The manuscript has been improved but there are some remaining issues that need to be addressed before acceptance, as outlined below:

Reviewer #3 has some remaining concerns that need to be addressed, including justification of the usage of GLM and description of the terms IC_75_, IC_90_ and IC_95_ and the way they were calculated.

*Reviewer #1:*

In their work studying the evolutionary stability of antibiotic collateral sensitivity interactions in *P. aeruginosa*, Barbosa and colleagues explored the effects of bacterial adaptation to antibiotic pairs in various combinations and treatment strategies. As a reviewer of a previous version of this manuscript, I found the topic of the paper as relevant and timely, the experiments well-designed and sufficiently novel, and the overall conclusions to be supported by the experimental data that was presented. My main criticism regarding the work in general was that I found the scope of the work somewhat limited based on the fact that the overwhelming majority of the data focused on a specific antibiotic pair and it was difficult to gauge how general these findings were when looking at a wider set of antibiotic pairs. In the revised version, the authors have clearly addressed these points by conducting an extensive set of additional experiments with other antibiotic pairs to get a much bigger picture regarding the general frequency of extinction, re-sensitization and multi-resistance with their experimental treatment. I think this adds value to their original findings and helps to put their original results with the CAR/GEN antibiotic pair into a clearer context.

*Reviewer #2:*

The authors have adequately addressed my concerns by including substantial new experimental work in their paper, and I think that this paper will make a good contribution to the growing literature on collateral sensitivity and antibiotic resistance.

*Reviewer #3:*

I have read "Evolutionary stability of collateral sensitivity to antibiotics in the model pathogen *Pseudomonas aeruginosa*" re-submitted to *eLife* following the initial round of peer-review by Barbosa and Roemhild et al. The authors have responded to all my initial comments and I largely find the responses and changes to the manuscript satisfactory.

Please find my comments to the added experiments below:

In my initial assessment I asked for more details concerning the extinct populations since the only evidence of "operating CS" was the initial susceptibility testing of ancestral clones founding these populations. As a response to this and other reviewers' comments the authors now included a new extensive set of additional experiments spanning 14 cases of reciprocal and non-reciprocal CS drug-pairs. The authors analyzed populations that did not go extinct and had developed reduced susceptibility to the second drug in the switch. They found that where single drug-treatment was followed by combinations of both drugs no re-sensitization was observed and thus lack of evolutionary stability. However, in mono-drug treatment the authors report 6 cases of re-sensitization towards the OLD drug suggesting that these drug-pairs displayed stable CS.

Data are now included from surviving populations that were excluded from the initial version and 2, potentially 3, cases of stable CS are reported from these populations which suggests that CS operates during evolution.

Six cases reported in text – only 5 are marked with asterisks in Figure 5 – STR/PIT combination significant too?

Using a generalized linear model, the authors then asked (1) which factors explained variation in extinction, and (2) drivers of resistance gains and re-sensitizations. They concluded from these analyses that (1) variation in extinction was significantly associated with the molecular target of the second drug, and (2) target of drug A, CS effect size, and growth rate all contributed to re-sensitization. Finally, the authors argue that resistance gains in unconstrained treatments was associated with initial effect size of CS and initial growth rates of resistant population.

There is no justification or description of the use of GLM in the text (this may be present in Figure 6—source data 1 frequently referred to).

Moreover, justification for factors affecting resistance gains seems to build on data presented in Figure 6C and 6D. In particular Figure 6D, the reporting tendency of co-variance seems unlikely to be significant?

Drug susceptibility are reported in terms of IC_75_, IC_90_, and IC_95_. I cannot see any description as to how these are estimated or the rationale for their use.

---

## [Author Response]

[Editors’ note: the author responses to the first round of peer review follow.]

Reviewer #1:[…]Essential revisions:1) The authors conclude that for the first pair of drugs investigated (STR-PIT) frequent extinctions (51% of the populations) during evolution was due to strong genetic constraints against an evolutionary response to overcome CS. Beyond the initial phenotype and number of extinct populations I find this claim largely unsubstantiated. In full awareness of how difficult it would be to include a "non-CS control experiment" it would in my opinion benefit the paper to know what actually happened in the extinct versus non-extinct populations. Could not alternative hypotheses be that reduced susceptibility (STR-PIT) require accumulation of more mutations than for the CAR-GEN pair or if classical synergy between STR and PIT for the combination exposures led to more extinctions (aminoglycosides and β-lactams are often synergetic, including in Pseudomonas)?

Thank you very much for this comment. As emphasized by the reviewer, it would be (i) interesting to better understand what exactly happened in the extinct populations, yet (ii) impossible to study this directly because of the unavailability of these extinct lineages for subsequent experimental analysis.

In the revised manuscript, we now decided to take a different approach to assess in more detail the factors which contribute to population extinction during collateral sensitivity switches: we performed a new set of evolution experiment, for which we challenged 38 independent populations, covering 14 distinct cases of collateral sensitivity, to overcome their collateral sensitivity (see also response to reviewer #2). This new experiment now allows us to assess statistically which factors favor population extinction. Such statistical analysis was not really possible with the previous data, which only included 4 cases of collateral sensitivity. In detail, the new experiments covered cases with a range of initial values for fitness costs, collateral sensitivity effect sizes, and different drug targets for either first or second antibiotic. We then used the evolution experiments to assess which factor(s) were significantly associated with extinction and resistance evolution. Based on the new results, extinction seems to be determined primarily by the order of drug targets during treatment (Figure 6A). As such, the new data is consistent with our previous observation that extinction is maximized in directional switches from aminoglycoside to β-lactam.

Consequently, we now conclude that extinctions are stochastic events that are affected by drug identity and order (see subsection “Repetition of experimental evolution revealed an influence of drug type on population extinction and drug re-sensitization” in the revised manuscript).

Moreover, synergism between drugs is unlikely to explain higher extinction rates since combinations with high and low extinction rates (STR+PIT and CAR+GEN) both display strong synergy (as quantified in our previous paper (Barbosa et al., 2018) and now more clearly indicated in subsection “Extinction rates were high even under mild selection regimes”). Moreover, reciprocal cases of collateral sensitivity, which consist of synergistic drug combinations, vary in their ability to overcome collateral sensitivity (i.e., a strong order-effect on extinction rates, as indicated for example in Figure 6A).

2) The authors demonstrate elegantly an example of negative epistasis between pleiotropic resistance mutations and this is an important contribution to our mechanistic understanding of CS/hypersensitivity where very little mechanistic details exist. However, since new treatment strategies are frequently mentioned throughout the manuscript, I believe the manuscript would benefit from a link to clinical reality. Do the observed mutations also emerge during therapy or are they specific for the experimental conditions? How does the observed resistance levels relate to existing clinical breakpoints?

Many thanks for this comment. We agree that this information was missing in the previous version of the manuscript. We have now included a short paragraph to the Discussion section on the prevalence of the observed genetic changes in clinical isolates of our pathogen.

3) The authors argue, and partially conclude, that if the costs of resistance are high, it is more difficult to overcome CS. If I understand correctly, this is based on Figure 3E,F and Supplementary Figure 2, where relative growth rates are used as a proxy for relative fitness before and after evolution. First, these experiments are done in antibiotic free media and not in conditions experienced during evolution. Second, would not the proper comparisons be with the PIT-STR pairs, by analyzing data from the evolved populations (that did not go extinct) as well as doing relative fitness measurements with the pairs of genetically re-constructed mutants to avoid bias from environmental adaptation?

We thank the reviewer for this important comment. We agree that in our assay, the fitness cost is measured in drug-free environments and thus conditions not experienced by the evolving bacteria during experimental evolution. We still find this measure useful because of the following two reasons. (1) It is a measure that at least indicates that the bacteria experienced some kind of general trade-off when adapting to the antibiotic environment. (2) It is a measure that is comparable across the quite distinct antibiotic treatments, thereby allowing us to identify more general adaptation trade-offs (i.e. different from collateral sensitivity). In order to avoid any misunderstanding, we now explain the relevance of this measure in more detail (subsection “Drug order determined re-sensitization or emergence of multidrug resistance”) and also refer to it as an adaptation trade-off rather than a direct fitness cost (legends to Figure 3 and Figure 6). Please note that we do assess fitness under the relevant drug conditions with our measures of population extinction, relative biomass, and drug resistance (see Figure 2, Figure 3 and Figure 5). Moreover, we now emphasize that in certain cases, reversal of drug resistance also ameliorates this second evolutionary trade-off (See Figure 3E and 3F, subsection “Drug order determined re-sensitization or emergence of multidrug resistance”).

We also thank the reviewer for suggesting experiments to broaden the general insight from our study. As already explained above, we now performed an additional set of evolution experiments for this purpose (rather than analysis of reconstructed mutants, which we did in our analysis of epistasis effects, Figure 4). Most interestingly, the detailed statistical analysis of the new data now indicates that the adaptation constraint is significantly associated with the bacteria’s evolutionary responses, especially their ability to gain resistance to the new drug B (see Figure 6D, statistical results in the text). This new insight is based on a larger set of drug conditions and thus collateral sensitivity switch types, thereby allowing us to draw careful conclusions on the generality of our findings. We integrated a careful discussion of these results into the revised Discussion section.

Reviewer #2:[…]My questions are the following:1) I found the scope of the work somewhat limited, as only two collateral-sensitive drug pairs were studied. It would be nice to see the same experimental setup on a greater number of reciprocal, collaterally-sensitive antibiotic pairs to get a better idea of the frequency of pairs that more frequently cause extinction and pairs that lead to resistance to the second antibiotic. From their previous work (Barbosa et al., 2017), this may be e.g. DOR/STR, CAR/PIT, CAR/STR; did the authors examine these strains?

We very much value this comment. As suggested, we have now performed a large validation experiment, in order to expand the scope of our study. In detail, we have performed a new set of evolution experiments, which cover a total of 14 distinct types of collateral sensitivity switches, which include several additional antibiotics, and which include 38 biologically independent replicate populations as starting material for the second step of experimental evolution. This new validation experiment now allows us to assess statistically which factors contribute significantly to evolutionary adaptation of the bacteria (for more details see subsection “Repetition of experimental evolution revealed an influence of drug type on population extinction and drug re-sensitization”). The new results are summarized in new Figure 5 and Figure 6. In general, we obtained similar results as with the clones. The importance of these findings are explained in the revised Discussion section.

2) On that note, it would have also been interesting to look at the resistance mechanisms of the survivors of the STR/PIT sequential treatments. The authors point out that this analysis was not possible because of the limited number of survivor populations that grew, however by my count, 31 out of 64 populations actually did survive from the various treatment regiments. It would be interesting to see if these populations are multidrug resistant or perhaps cases of resensitization can be observed here as well.

We appreciate this comment. We have now included equivalent data to Figures 3A-D for the STR/PIT treatment (Figure 3—figure supplement 1). We do see similar patterns of resensitization or the evolution of multidrug resistance. The patterns are less clear for this drug due to stronger between-replicate variation for the more potent switch (STR>PIT), where some replicates showed clear re-sensitization but others the opposite. Such dynamics could potentially be explained by a larger diversity of accessible adaptive mutations. However, its detailed analysis remains hampered by the fact that for certain treatment combinations, only few replicates are available for analysis. Therefore, we found it more reliable to focus the genomic analysis on the drug switch, for which a larger and statistically relevant number of replicates is still available and which may then serve as a test case to explore what kind of mechanisms may help the bacteria to overcome collateral sensitivity. Our genetic analysis did indeed allow us to identify and characterize such mechanisms, including new insights into the importance of epistatic interactions. We consider these findings novel and of relevance for our understanding of evolved collateral sensitivity.

In order to broaden our perspective, we then decided to rather focus on performance of a validation evolution experiment. This validation experiment allowed us to statistically assess the factors that minimize counteradaptations against collateral sensitivity, which was the main focus of our work.

3) I found it interesting that the PIT/STR combination that seems to be the most effective in this study did not show reciprocal collateral sensitivity in the majority of cases in the authors' previous work (Barbosa et al., 2017). Only one out of 10 STR adapted lines was hypersensitive to PIT while cross-resistance arose in 5 of the cases. Can the fact that cross-resistance seems to be more common with this pair in the STR -> PIT direction mean that this combination would actually not be a prudent choice for treatment against P. aeruginosa PA14?

We thank the reviewer for this important observation. We now attempted to address this comment with the help of our new validation experiment, which allowed us to statistically evaluate which factors and which drug properties favor desired treatment outcomes (e.g., extinction, reduced levels of multidrug resistance). Our analysis now suggests that, in general, starting with aminoglycoside antibiotics and switching to β-lactam will be more effective than the reverse switch or switches based on other drug types. See also above replies. We still agree with the reviewer that variation in the evolution of collateral sensitivity in response to treatment with the initially applied drug is similarly of key importance. To accommodate this point, we now re-wrote the final paragraph of the discussion. Here we now conclude that: “The success of this treatment strategy depends on several key factors. One critical prerequisite is that collateral sensitivity does evolve in response to treatment, which may vary depending on alternative evolutionary paths to resistance against the initially used drug A (Barbosa et al., 2017; Nichol et al., 2019). Our new data now suggests that treatment can be further optimized by switching from an aminoglycoside to a β-lactam and/or by focusing on drugs, for which resistance is associated with large general adaptation trade-offs, and/or by using drug combinations with small collateral sensitivity effect sizes.” See Discussion section.

Reviewer #3:In this manuscript, the authors use experimental evolution to investigate the consequences of collateral sensitivities that evolved when Pseudomonas aeruginosa evolved resistance to antibiotics in a previously published selection experiment. This paper investigates a hot topic in evolutionary biology (antibiotic resistance) using previously established methods, and this paper clearly makes a novel contribution to our understanding of trade-offs between resistance to alternative antibiotics (collateral sensitivity). The genome sequencing and mutant reconstruction provide good support for the experimental evolution component of the paper, and the use of this approach adds an interesting layer of detail to the paper. However, I have some reservations about the experimental design and data analysis for this paper, and I question the breadth of conclusions that can be drawn from this study.1) Experimental design and possible pseudo-replication:The authors setup their selection experiment with 2 replicate cultures of 4 clones from 4 different populations that were previously selected for resistance to CAR, GEN, STR and PIP. I have two criticisms of this design.a) In their analysis of extinction rates and evolutionary dynamics (Figure 2 and Figure 3), the authors treat the 2 replicates of each clone as being independent of each other, even though they were derived from the same clone. It is difficult to evaluate the impact of this pseudo-replication on the results of the analysis, and this is an issue that the authors need to address.b) By choosing multiple clones from a single source population, the authors limit the generality of their findings. For example, I do not think that it is a fair conclusion to say that PIT adapted populations tend to go extinct when they are challenged with STR (and vice versa), because all of the PIT adapted clones came from a single population that evolved PIT resistance.c) Overall, I think that the study would have been stronger if the authors had instead started their selection experiments with 8 independently evolved clones, each from a different source population.

Many thanks for comments 1a, b and c. In the revised manuscript, we now tackled the criticism of re-sampling from the same source population by performing a new, large-scale validation experiment that investigates the stability of collateral sensitivity starting with 6 independently evolved populations (Figure 5). As for possible pseudo-replication, we fully agree that it is more appropriate to treat the two experimental lineages that were derived from a single clone as technical replicates. This was already part of our previous GLM analysis. Many apologies if this was not clear.

2 Measurements in resistance:Clinical microbiologists measure antibiotic resistance using standardized techniques to estimate MIC. The authors measure changes in the resistance of evolved populations by measuring changes in the area under the inhibition curve (Figure 3). This data provides a more subtle and quantitative measure of resistance, but this metric lacks a unit and the authors should also measure changes in MIC so that their results can be placed more easily in the context of antibiotic resistance. For example, it is unclear to what extent adaptation to new antibiotics changed resistance to old antibiotics. This data will help clinicians to evaluate the extent to which evolutionary changes in resistance are likely to lead to different outcomes in the clinic.

We thank the reviewer for this this helpful suggestion. We have now included the IC_90_ fold change for the two datasets in Supplementary file 1—figure 3—supplementary table 3 and Supplementary file 1—Figure 5—supplementary table 2. The results are consistent with our previous findings. Moreover, we also performed the statistical analysis for the new data using the different alternative measures and similarly obtained consistent results (see Supplementary file 1—Figure 6—supplementary table 1 and Supplementary file 1—Figure 6—supplementary table 2, Figure 6—source data 1 and Figure 6—source data 2 and subsection “Repetition of experimental evolution revealed an influence of drug type on population extinction and drug re-sensitization”).

3) Applicability of findings:The authors conclude that the findings of their study could help to guide the design of sustainable antibiotic therapies in the clinic (Discussion section). I am not convinced that this is the case, and I think that the authors should be careful about overstating the applied significance of their work. Because the authors used only a single source population for each selection treatment, they cannot make any general conclusions about how different antibiotics collateral sensitivity (i.e. this study does not support the idea that PIT followed by STR will be a generally effective strategy for treating Pseudomonas infections). A further limitation to the applicability of this study comes from the fact that both collateral sensitivity combinations relied on using aminoglycoside antibiotics (STR and GEN); these antibiotics are rarely used to treat Pseudomonas infections (i.e. https://www.thoracic.org/statements/resources/tb-opi/hap-vap-guidelines-2016.pdf). This is a minor point, but I think that it is important to be careful about how researchers in this area interpret their findings in the context of clinical use of antibiotics.

We appreciate this comment. We admit that the conclusions in our previous manuscript were not sufficiently cautious. In response to this comment, we now rephrased the relevant parts of our discussion and tried to be more careful in our conclusions on the clinical applicability of our findings. Importantly, our results highlight that the concept of collateral sensitivity cannot be applied in a simple form to patient treatment. Instead, it appears that certain directions of collateral sensitivity (i.e., aminoglycoside followed by a βlactam) are more robust against bacterial counteradaptation. Here, it is important that this conclusion is strongly supported by our new validation experiment (Figure 6), in which we included (i) a larger number of different types of collateral sensitivity switches, (ii) a larger number of antibiotics, and (iii) a total of 38 independent replicate populations of evolved resistances. Considering that evolution of collateral sensitivity as the first step is not always predictable (see Barbosa et al., 2017), its use for treatment requires more detailed patient-specific evaluations. These are important, clinically relevant insights, which clearly warrant further exploration, as we now emphasize in our final conclusions (Discussion section). Moreover, we also agree that aminoglycosides are often not used to treat *P. aeruginosa* infections. However, sometimes, they still are, depending on country, exact infection type, patient history, hospital stewardship, and other factors. In our personal experience, we have repeatedly observed the use of the aminoglycoside tobramycin in the cystic fibrosis clinic in Kiel (see for example Tueffers et al., 2019). Therefore, in principle, our specific results on aminoglycosides can help those treatments, where these drugs are an option, and therefore we consider our conclusions to be valid and clinically relevant.

[Editors' note: the author responses to the re-review follow.]

Reviewer #3 has some remaining concerns that need to be addressed, including justification of the usage of GLM and description of the terms IC_75_, IC_90_ and IC_95_ and the way they were calculated.Reviewer #3:I have read "Evolutionary stability of collateral sensitivity to antibiotics in the model pathogen Pseudomonas aeruginosa" re-submitted to eLife following the initial round of peer-review by Barbosa and Roemhild et al. The authors have responded to all my initial comments and I largely find the responses and changes to the manuscript satisfactory.Please find my comments to the added experiments below:In my initial assessment I asked for more details concerning the extinct populations since the only evidence of "operating CS" was the initial susceptibility testing of ancestral clones founding these populations. As a response to this and other reviewers' comments the authors now included a new extensive set of additional experiments spanning 14 cases of reciprocal and non-reciprocal CS drug-pairs. The authors analyzed populations that did not go extinct and had developed reduced susceptibility to the second drug in the switch. They found that where single drug-treatment was followed by combinations of both drugs no re-sensitization was observed and thus lack of evolutionary stability. However, in mono-drug treatment the authors report 6 cases of re-sensitization towards the OLD drug suggesting that these drug-pairs displayed stable CS.Data are now included from surviving populations that were excluded from the initial version and 2, potentially 3, cases of stable CS are reported from these populations which suggests that CS operates during evolution.Six cases reported in text – only 5 are marked with asterisks in Figure 5 – STR/PIT combination significant too?

Many thanks for this observation. Our count of “6 of the 14 cases” includes the 5 cases marked by asterisks in Figure 5 for which a statistical evaluation was performed, and the case of STR>PIT. The latter only had a single replicate, and we therefore found a statistical evaluation inappropriate. However, this replicate showed very strong resensitization, such that we included it in the above count. Nevertheless, statistical significance could only be inferred for 5 cases. We have now clarified this point in subsection “Repetition of experimental evolution revealed an influence of drug type on population.

Using a generalized linear model, the authors then asked (1) which factors explained variation in extinction, and (2) drivers of resistance gains and re-sensitizations. They concluded from these analyses that (1) variation in extinction was significantly associated with the molecular target of the second drug, and (2) target of drug A, CS effect size, and growth rate all contributed to re-sensitization. Finally, the authors argue that resistance gains in unconstrained treatments was associated with initial effect size of CS and initial growth rates of resistant population.There is no justification or description of the use of GLM in the text (this may be present in Figure 6—source data 1 frequently referred to).Moreover, justification for factors affecting resistance gains seems to build on data presented in Figure 6C and 6D. In particular Figure 6D, the reporting tendency of co-variance seems unlikely to be significant?

Many thanks for these two comments. We have now revised the methods section and included a description and justification of the GLM analyses subsection, “Statistical analysis for association with evolutionary stability”. Please note that the association shown in Figure 6D was found to be significant, as inferred from GLM analysis reported in Supplementary file 1. In order to avoid misunderstandings, we now replaced “covariation”/“covaried” by the more correct terms “significant association”/“significantly associated” in the legend to Figure 6, because in a strict sense our GLM analysis only tested for associations. Moreover, we also removed any reference to correlation analysis in this context, because – as just emphasized – the associations were assessed using GLM analysis. See adjustments in subsection “Repetition of experimental evolution revealed an influence of drug type on population extinction and drug re-sensitization” and the legend to Figure 6.

Drug susceptibility are reported in terms of IC_75_, IC_90_, and IC_95_. I cannot see any description as to how these are estimated or the rationale for their use.

Many thanks for bringing this to our attention. In the revised manuscript, we now describe these terms and the methods of quantification in subsection “Measurements of reciprocal collateral sensitivity”. In general, the metrics IC_75_, IC_90_, and IC_95_ are very strongly correlated. IC_95_ is a close approximation of the MIC. High IC values are technically hard to estimate, when the highest concentrations do not result in total inhibition, especially when using OD measurements for their inference (as was the case in our study), which are usually subject to unfavorable signal-to-noise ratios close to the zero OD measurements. As such high levels of noise were observed for two samples in our validation data set, we instead chose to work with IC_75_ which could be accurately determined for all samples.